**Debris cover effects on energy and mass balance of Batura Glacier in the Karakoram over the**
**past 20 years**
Yu Zhu[1,2], Shiyin Liu[1,2,5*], Ben W. Brock[3], Lide Tian[1,2], Ying Yi[1,2], Fuming Xie[1,2], Donghui Shangguan[4], and
YiYuan Shen[1,2]
[1] Yunnan Key Laboratory of International Rivers and Transboundary Eco-security, 650091 Kunming, China
[2] Institute of International Rivers and Eco-Security, Yunnan University, 650091 Kunming, China
[3] Department of Geography and Environmental Sciences, Northumbria University, Newcastle upon Tyne, NE1 8ST,
UK
[4] Northwest Institute of Eco-Environment and Resources, Chinese Academy of Sciences, Lanzhou 730000, China
[5]International Joint Laboratory of China-Laos-Bangladesh-Myanmar Natural Resources Remote Sensing
Monitoring
Corresponding author: Shiyin Liu (shiyin.liu@ynu.edu.cn)
Abstract:
The influence of supraglacial debris cover on glacier mass balance in the Karakoram is noteworthy. However,
understanding of how debris cover affects the seasonal and long-term variations in glacier mass balance through
alterations in the glacier's energy budget is incomplete. The present study coupled an energy-mass balance model
with heat conduction within debris layers on debris-covered Batura Glacier in Hunza valley, to demonstrate the
influence of debris cover on glacial surface energy and mass exchanges during 2000-2020. The mass balance of
Batura Glacier is estimated to be -0.262 ± 0.561 m w.e. yr$^{-1}$, with debris cover accounting for a 45% reduction in
the negative mass balance. Due to the presence of debris cover, a significant portion of incoming energy is utilized
for heating debris, leading to a large energy emission to atmosphere via thermal radiation and turbulent sensible
heat. This, in turn, reduces the melt latent heat energy at the glacier surface. We found that the mass balance exhibits
a pronounced arch-shaped structure along the elevation gradient, which is associated with the distribution of debris
thickness and the increasing impact of debris cover on the energy budget with decreasing elevation. Through a
comprehensive analysis of the energy transfer within each debris layer, we have demonstrated that the primary
impact of debris cover lies in its ability to modify the energy flux reaching the surface of the glacier. Thicker debris
cover results in a smaller temperature gradient within debris layers, consequently reducing energy reaching the
debris-ice interface. Over the past two decades, Batura Glacier exhibited a trend toward less negative mass balance,
likely linked to a decrease in air temperature and reduced ablation in areas with thin or sparse debris cover.

1 Introduction

Karakoram Glaciers have maintained a relative stable status under atmospheric warming, compared with other High Mountain Asia (HMA) glaciers over past 30 years (Zemp et al., 2019; Nie et al., 2021; Gardelle et al., 2012), a phenomenon which has been referred to as the "Karakoram Anomaly" (Hewitt, 2005). However, due to the influence of topographical and supraglacial features, the rate of glacier change across this region exhibits a distinct spatial heterogeneity. Notably, supraglacial debris plays a key role in mass change on many glaciers in the Karakoram. Over the past three decades, a discernible expansion of supraglacial debris has been observed throughout the Karakoram region (Xie et al., 2023), achieving a notable coverage of 21% in areas such as the Hunza river basin (Xie et al., 2020). Ever since Hewitt (2005) identified the inhibitory effect of supraglacial debris on melt, particularly below 3500m, as a possible explanation for the "Karakoram Anomaly", mapping the changes in the extent and mass changes of debris-covered glaciers has been the focus of several recent studies (e.g., Mölg et al. (2018), Azam et al. (2018), Xie et al. (2020)).

Until now, the direct assessment of debris impact on Karakoram glaciers has been limited to a few glaciological measurements conducted over short periods. Mihalcea et al. (2008) modeled debris-covered ice ablation across the ablation area of the Baltoro glacier, employing a distributed approach that calculated conductive heat flux through the debris layer. However, their study lacked a thorough analysis of the debris effect on ice melt. Recently, Huo et al. (2021b) conducted advanced research on the Baltoro glacier, presenting a model that comprehensively characterizes ablation dynamics, considering temporally-linked radiative forcing, surface geomorphological evolution, and gravitational debris flux. They emphasized the role of system couplings and feedbacks between surface morphology, melt, and debris transport, revealing an overall increase in ablation due to high-frequency topographic variations leading to a larger area with thin debris cover. At a larger scale, such as the Central Karakoram, Minora et al. (2015) reported a noticeable difference in melt rates between debris-covered and debris-free ice, utilizing an enhanced temperature index model. Furthermore, by conducting a comparative modelling study of ice melt with and without debris cover for one ablation season in 2004, Collier et al. (2015) estimated that debris cover reduced ablation by approximately 14% in the Karakoram. They attributed this significant reduction to insulation by thick debris cover exceeding increases in melt under thin debris. Additionally, Groos et al. (2017) confirmed that debris influences the anomalous behavior of glaciers in the Karakoram using a surface mass balance model. They emphasized that debris is not the sole driver, however; factors such as favorable meteorological conditions and the timing of the main precipitation season also contribute. Consequently, the distribution of debris

holds strong potential for affecting atmosphere–glacier feedbacks and glacier ablation in this region, warranting more comprehensive exploration of the intricate dynamics of mass and heat exchange within the debris in the Karakoram.

Supraglacial debris up to a few centimeters thickness generally increases melt due to lowered albedo and increased heat absorption at the surface (Collier et al., 2014), while thicker debris cover typically suppresses the melt rate through insulation (Østrem, 1959; Nicholson and Benn, 2006; Bisset et al., 2020). These contrasting effects have been demonstrated by many recent studies (Gardelle et al., 2012; Nuimura et al., 2017; Basnett et al., 2013; Fujita and Sakai, 2014). The reduction of ablation associated with increasing debris thickness down glacier can lead to an inverted mass-balance elevation profile on the debris-covered ablation zone, which has profound implications on the evolution of a glacier under a warming climate (Banerjee, 2017). Some field studies have also identified diverse effects on melt rates of debris cover with different thickness in Karakoram; one particular finding showed that thin debris cover, e.g. 0.5 cm in thickness, does not accelerate ice melting in this region (Muhammad et al., 2020). However, some remote sensing based research proposed that while thick debris typically inhibits the melt rate, the overall ablation on glaciers extensively covered in debris is still significant (Kääb et al., 2012). These findings imply that understanding of the process and feedback mechanisms governing ablation of debris-covered glaciers in this region is still incomplete. Therefore, it is important to quantify not only the amplitude of melt under time-variable debris cover but also its role in the "Karakoram Anomaly" by assessing the thermal properties of debris layers of different thickness.

Field glaciological and meteorological observations on glaciers in the Karakoram are limited by logistical and political constraints (Mayer et al., 2014; Mihalcea et al., 2008). Consequently, a significant knowledge gap exists for debris thickness and its thermal properties as well as the complex coupling of meteorology with heat exchange over glaciers and in debris layers. A limited number of previous melt process investigations under debris layers, e.g., Juen et al. (2014), Evatt et al. (2015), Muhammad et al. (2020), supported by remote sensing observations and climate reanalysis data, have enabled physically-based numerical modeling to provide insight into thermal dynamics within supraglacial debris. For example, Huo et al. (2021b) provided new insights into the relationships between ablation dynamics, surface morphology and debris transport, while Collier et al. (2015) developed understanding of how debris cover affects the atmosphere–glacier feedback processes during the melt season. However, despite these advancements, certain aspects remain insufficiently addressed. Specifically, the seasonal variations and long-term changes in melt patterns, along with the manner in which debris cover exerts its influence on such variations, have

not been comprehensively studied. Understanding these dynamics is essential not only for establishing the physical basis of the "Karakoram Anomaly" but also for quantifying the extent to which debris cover contributes to this phenomenon. In this study, we applied an energy-mass balance model coupled with heat conduction within debris layers on Batura Glacier in Hunza valley, Karakoram to demonstrate the influence of debris cover on glacial melt. We aim to: (1) reconstruct the long-term mass balance history of the Batura Glacier, a representative debris-covered glacier in the region; and (2) numerically estimate the distributed ice melt rate under the spatially-heterogeneous supraglacial debris of the Batura Glacier. By enhancing our understanding of glacier mass balance behavior and its relationship to debris cover energy budgets in the Karakoram over the last two decades, this research adds significantly to existing knowledge in this field.

2 Study site

The Batura Glacier, located in northwest Karakoram, stands as one of the most prodigious valley-type glaciers in the lower latitudes, extending over a length of more than 50 km and encompassing an expansive area exceeding 310 km$^2$ (Xie et al., 2023) (Figure 1). Approximately 24% (~76 km$^2$) of the glacier's area is covered with debris (Xie et al., 2023), while its thickness in the part below 3000 m a.s.l. surpasses 50 cm (Gao et al., 2020). Due to the heavy debris cover, Batura Glacier presents a hummocky topography and a concave longitudinal surface profile. Because of the large difference in density between ice and debris, the heavily debris-covered glacier section has higher hydrostatic pressure at the glacier bottom (Gao et al., 2020).. Influenced by the prevailing Westerlies, the Batura Glacier receives abundant snowfall (exceeding 1000 mm w.e. at altitudes above 5000 meters) in the high-altitude region (Lanzhou Institute of Glaciology and Geocryology, 1980). In addition, the interaction of the South Asian monsoon and Karakoram vortex cause localised cooling over Karakoram, leading to a low air temperature in summer (Dimri, 2021; Forsythe et al., 2017). As observed by (Lanzhou Institute of Glaciology and Geocryology, 1980), the Batura glacier is characterized by a relatively low average annual air temperature compared to observed glaciers in Tianshan and Himalayas, particularly near the annual snowline, where temperatures close to, or below, 0 °C endure throughout the year, averaging approximately -5°C annually. The glacier displays a rapid flow velocity, with a maximum rate reaching up to 517.5 m yr$^{-1}$, facilitated by a high rate of mass turnover, and undergoes frequent periods of advance and retreat, while remaining devoid of any surging events (Bhambri et al., 2017).

Since the comprehensive investigation on Batura Glacier conducted by Lanzhou Institute of Glaciology and Geocryology during 1974-1975, there has been a scarcity of systematic observations and studies on this glacier.

Contemporary investigations of Batura Glacier primarily utilize remote sensing observations, focusing on the glacier dynamics and long-term mass balance, e.g. Rankl and Braun (2016), Wu et al. (2021). There is a challenge in understanding glacier ablation, associated secondary hazards such as glacier floods, and the contribution of glacier runoff to river replenishment.

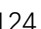

**Figure 1** Study area. (a) Image of Batura Glacier in 2019 (Synthesized using Sentinel-2 data). (b) Geographic location of Batura Glacier, with the red line outlining the Karakoram, and the blue line outlining the Hunza valley within which Batura Glacier is situated. The three weather stations labeled are Khunjerab, Ziarat, and Naltar. (c) Surface topography of Batura Glacier. (d) Measurement profiles of debris thickness.

3 Data and methods
3.1 Data
3.1.1 Observations
An automatic weather station (AWS 1, 74.661° E, 36.550° N, 3390 m) was set up at Batura Glacier on 23
September 2013 by the Northwest Institute of Eco-Environment and Resources, Chinese Academy of Sciences
(Figure 1a) and has been in continuous operation since then (Figure 1a). Meteorological variables observed at the
station are maximum/minimum wind speed and direction, maximum/minimum air temperature, relative humidity,
atmospheric pressure, upward and downward long- and shortwave radiations and precipitation, recorded on a daily
basis. In this study, we use data from AWS1 in the period 23 September 2013 to 9 May 2018 for the bias correction
of HAR v2 (High Asia Refined) reanalysis data (Wang et al., 2020) (see section 3.1.2) and for the accuracy
assessment of the energy and mass balance simulations. The second AWS (AWS 2, 74.851° E, 36.506° N, 2664 m)
was set up in August 2019 by Yunnan University on a debris-covered part of the tongue of the Batura Glacier. The
AWS2 records the same climatic factors as AWS1, but it doesn't measure precipitation. We use data from AWS 2
between 1 September 2019 to 25 November 2020 to evaluate the reliability of parameters for energy balance in the
debris-covered area. The technical specifications for the sensors used in both AWSs are detailed in Table S1. We
additionally used daily maximum/minimum temperatures and precipitation from stations at Khunjerab, Ziarat, and
Naltar in the Hunza Valley (Figure 1b) covering the period from January 1, 1999 to December 31, 2008, provided
by Water and Power Development Authority (WAPDA), Pakistan, to assess the accuracy of HAR in the Hunza basin.
The debris thickness at the terminus of the Batura Glacier (2014) was surveyed by WAPDA and provided by a
research group at COMSATS University Islamabad of Pakistan. Additionally, we collected measurements of debris
thickness at six sample points near AWS 2 during fieldwork in 2019.

3.1.2 Reanalysis data
The HAR reanalysis data is a product derived from the dynamical downscaling process using the Weather
Research and Forecasting (WRF) model. The driving data for the first version is FNL (Final) Operational Global
Analysis data, while the second version uses ERA5-atmospheric (0.25°) data (Wang et al., 2020). Compared to the
first version, the second version expanded the spatial range of the simulation and extended the time range and will
continue to receive updates (see Wang et al. (2020)). In the production of the meteorological variables, the dynamic
assimilation of downscaled results was achieved using satellite products and ground observations such as wind
speed, wind direction, temperature, and geopotential height. This process significantly improved the accuracy and
credibility of the downscaling simulation. Notably, the HAR product has shown great potential in reflecting regional
water vapor transport processes (Curio et al., 2015) as well as spatial heterogeneity and seasonal variations in
precipitation and temperature (Maussion et al., 2014).
3.1.3 Other data
The geodetic mass balances for Batura Glacier generated by Brun et al. (2017), Wu et al. (2020), Shean et al.
(2020), and Hugonnet et al. (2021) were utilized to validate the energy and mass balance simulation results. These
mass balance data were derived from elevation differences with some assumptions such as ice density, etc. With the
exception of the five-year mass balance (2000-2020) produced by Hugonnet et al. (2021), the other data only show
the long-term mass balance status after 2000. Time ranges for all mass balance data can be found in Figure 3. The
30 m resolution DEM from the Shuttle Radar Topography Mission (SRTM) was used to generate required terrain
factors, while the glacier boundary was defined using the most recent delineation published by Xie et al. (2023).
3.2 Methods
3.2.1 The physically-based energy-mass balance (EMB) model
The EMB model for snow and ice is a distributed model that combines surface energy processes with a
subsurface evolution scheme for snow and ice (COSIPY v1.3) which was developed by Sauter et al. (2020). Details
of the model relating to applied parametrizations, physical principles and technical infrastructure have been
described in Huintjes et al. (2015b), Sauter et al. (2020) and (Arndt and Schneider, 2023). In common with previous
energy balance models, the surface energy budget is defined as the sum of the net radiation, turbulent heat fluxes
(including sensible heat flux $q_{sh}$ and latent heat flux $q_{lh}$), conductive heat flux ($q_g$), sensible heat flux of rain ($q_{rr}$)
and melt energy ($q_{me}$) (Eq.1). The net radiation is the sum of the net shortwave radiation calculated from incoming
shortwave radiation ($q_{sw_{in}}$) and surface albedo ($\alpha$), incoming longwave radiation ($q_{lw_{in}}$) and outcoming longwave
radiation ($q_{lw_{out}}$). To link the surface energy balance to subsurface thermal conduction, the snow/ice surface
temperature ($T_{s\_si}$) is defined as an upper Neumann boundary condition. The penetrating scheme of shortwave
radiation is based on Bintanja and Van (1995).

$$q_{me} = q_{sw_{in}}(1 - \alpha) + q_{lw_{in}} + q_{lw_{out}} + q_{sh} + q_{lh} + q_{rr} + q_g \qquad (1)$$

The glacier melt is solved using $q_{me}$ and penetrating shortwave radiation, while the sublimation is solved
using $q_{lh}$. Combined with the snowfall and refreezing of meltwater (or rain), the total mass balance of the glacier
surface can be calculated (Eq.2). The sum of subsurface melt ($m_{sub}$) due to penetrating shortwave radiation energy
and the refreezing of meltwater (or rain) (refreeze), is defined as the internal mass balance. The internal ablation
occurs when temperature at a specific layer reaches the melting temperature ($T_m$). Internal meltwater, in combination
with infiltrated surface meltwater, can be stored in the snow layers. Once a layer becomes saturated, meltwater will
drain into the next layer until the liquid water content within all layers is less than a defined ratio, or else the
meltwater runs off when it reaches the lowest model layer. In this process, a part of the meltwater refreezes when
the temperature at a layer is less than $T_m$. Full details for resolving mass and energy budgets in the EMB can be
found in Sauter et al. (2020).

$$mb = \left(\frac{q_{me}}{L_m} + \frac{q_{lh}}{L_v} + \text{snowfall}\right) + (m_{sub} + \text{refreeze}) \tag{2}$$

where $L_m$ is the latent heat of ice melt and $L_v$ is the latent heat of sublimation or condensation.
The debris energy balance is calculated according to the model of Reid and Brock (2010), and the reader is
referred to their paper for a detailed description of the model. The sum of energy fluxes at the surface is essentially
the same as Eq. 1, but because debris does not melt, the debris surface temperature ($T_{s\_d}$) is assumed to change such
that these fluxes sum to zero:

$$q_{sw_{in}}(1 - \alpha) + q_{lw_{in}}(T_{s\_d}) + q_{lw_{out}}(T_{s\_d}) + q_{sh}(T_{s\_d}) + q_{lh}(T_{s\_d}) + q_{rr}(T_{s\_d}) + q_g(T_{s\_d}) = 0 \tag{3}$$

The circularity in solving for $T_{s\_d}$ is resolved using a numerical Newton-Raphson method (Eq. 4). Conduction
through the debris is then calculated using a Crank-Nicholson scheme with intermediate temperature layers for a
set depth, and boundary conditions determined by the newly calculated $T_{s\_d}$ and the temperature at the debris-ice
interface, which is assumed to stay at zero (Eq. 5). The ablation rate is determined from the conductive heat flux to
the first (uppermost) ice layer, found using the temperature gradient between the lowest debris layer and the ice (Eq.
6). The detailed solution processes for Eq. 4~6 can be found in Figure 2 and Appendix materials in Reid and Brock

208 (2010).

$$T_{s\_d}(n + 1) = T_{s\_d}(n) - \frac{fun(T_s(n))}{fun'(T_s(n))'} \tag{4}$$

where, $T_{s\_d}(n)$ and $fun(T_{s\_d}(n))$ refer to the temperature and the total energy flux at nth debris layer. The
termination condition for this solution is set as $T_s(n + 1) - T_s(n) < 0.01$.

$$q_G = -k_d \left(\frac{dT_s}{dz}\right) \approx k_d \frac{T_{s\_d}(N-1) - T_m}{h} \tag{5}$$

$$Melt_{deb} = \frac{q_G}{\rho_i L_f} \tag{6}$$

where, $h$ represents the thickness of each layer, $n$ represents nth debris layer, $N$ represents the number of
calculation layers, $T_m$ represents the melting temperature of ice, and $k_d$ is the thermal conductivity of
supraglacial debris. $Melt_{deb}$ refers to the ablation rate of ice at the debris interface.

217        In the model run, the initialization of the model was firstly conducted using the defined parameters. The most

important in this step was the establishment of the temperature profile, which was initialized with air temperature
($T_a$) and bottom temperature ($T_b$) by using linear interpolation. The second step involved recalculating the
temperature profile, involving two scenarios: (1) In debris-free areas, the temperature profile was calculated entirely
according to the COSIPY. Initially, the temperature profile was computed without considering the impacts of
refreezing or subsurface melt but factoring in temperature increase due to penetrating radiation. If a snow/firn pack
is present, the densification of the dry snow pack was calculated using an empirical relation (Herron and Langway,
1980). After densification, the available surface and subsurface meltwater percolated downward, with a small
amount retained in each layer. Subsequently, the temperature changes resulting from refreezing of meltwater were
computed, updating the subsurface layer temperature. In debris-covered areas, when snow presented, the snow-
debris interface temperature was first obtained using the snow layer temperature update scheme of the COSIPY
model. This temperature was then set as the debris surface temperature. By defining the debris-ice interface
temperature as zero, the debris layer temperature was then calculated using Eq. 5. In the absence of snow, the model
employs the debris layer temperature update scheme described by Reid and Brock (2010). The third step involved
using the surface temperature obtained from the second step, combined with glacier surface meteorological
parameters, to calculate the surface energy balance and surface melt. The primary physical processes of the model
are illustrated in Figure 2. In this study, a two-year spin-up was implemented to allow the model to adapt to the
surrounding conditions (Huintjes et al., 2015a).

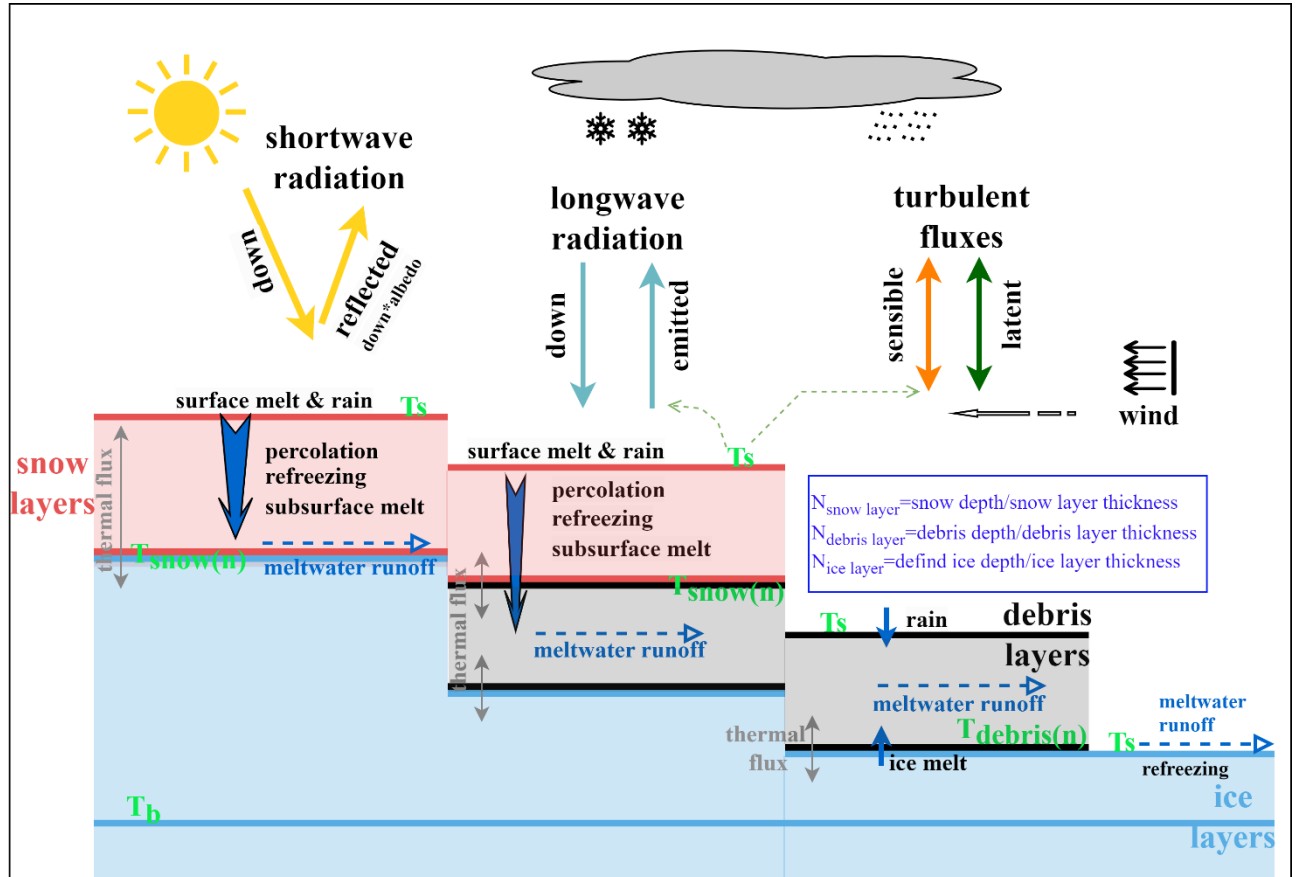

**Figure 2** General scheme of the model used in the current study with fluxes and physical processes. $T_s$ represents surface temperature, solved for using the heat conduction equation. The solution process varies depending on the different surface cover conditions of the glacier. $T_s$ is a crucial variable linking the energy exchange between the glacier and the atmosphere. $T_s$ is primarily used to calculate sensible heat flux and emitted longwave radiation. Reflected shortwave radiation is mainly determined by surface albedo. In the case of snow cover, the albedo changes continuously with snowmelt and densification. $T_{snow(n)}$ represents the temperature of the nth snow layer, reflecting the energy flux at the snow-ice interface or snow-debris interface. $T_{debris(n)}$ represents the temperature of the nth debris layer, reflecting the energy flux at the debris-ice interface. These two variables are important for characterizing the internal energy balance of the glacier.

In the model, the layers of snow, debris, and ice were dynamically calculated based on their individually specified thicknesses. Considering that the temperature of the ice layer does not change with increasing thickness below a certain depth in glaciers, a depth of 10 m for the ice layer was set, following Huintjes (2014). As ice temperature cannot exceed 0 ℃, the boundary conditions at snow-debris interfaces were configured similarly, following an analogous scenario that the temperature of snow-debris interface remains below 0 ℃ (Giese et al., 2020). Based on this, we made the assumption that any rain or snowmelt water does not refreeze within the debris

layer, and the infiltration of such water does not alter the temperature of the debris layer. The temperature boundary
condition at the debris-ice interface follows Reid and Brock (2010), ensuring that the temperature of debris-ice
interfaces remains below 0 ℃. For the lower boundary condition (bottom temperature), values referenced from
Huintjes (2014) are employed, derived from observational data. To prevent ice layer temperatures from exceeding
freezing level, a heating mechanism is applied to the ice layer above the bottom layer, directing above-freezing
energy into the melting process.
In this study, the model simulations were conducted using a high-performance server, equipped with dual
Intel Xeon CPU E5-2687W processors (48 threads), 768 GB of RAM, and dual Quadro P6000 (24G) GPUs for
acceleration. We conducted simulations that compared scenarios with and without supraglacial debris on the Batura
Glacier to assess the influence of debris on mass balance.
3.2.2 Model setup and input data
In this study, HAR v2 data were used to drive the model to simulate the energy and mass balance of the Batura
Glacier from 2000 to 2020. The meteorological variables in HAR v2 selected to meet the requirements of the energy
balance simulation include precipitation, air temperature at 2 m, wind speed (*u*- and *v*- components at 10 m),
atmospheric pressure, specific humidity, downward shortwave radiation, and cloud cover. The 10 m wind speed was
converted to 2 m using an empirical formula provided by Allen et al. (1998), while specific humidity was converted
to relative humidity using the formula given by Bolton (1980) utilizing the 2 m air temperature and atmospheric
pressure. Air temperature was calibrated at the basin scale using a gridded bias factor. The gridded bias was
interpolated by the nearest-neighbor method, with the bias at each station calculated between the observed and HAR
temperatures. After correction, a small bias range of ±1℃ was observed between HAR temperature and station
temperature, with a Pearson correlation coefficient of 0.98. Details regarding the precipitation calibration can be
found in Appendix A1. Due to lack of observations for other variables, no further validation before statistical
downscaling was conducted at the basin scale in this study. However, minor adjustments were applied for
downscaled other variables. These adjustments were made using scale factors calculated through the least squares
method, considering the downscaled results and observed values at the two stations on Batura glacier.
We utilized the data from Rounce et al. (2021) based on an inversed energy balance modeling procedure to
calculate debris thickness inputs. The debris thickness with a 100 m resolution is resampled to 300 m using an
inverse distance weighted interpolation method to match the simulation resolution. We validated the simulated
debris thickness using observed data, which showed an average deviation of 6 cm. However, it should be noted that

the Rounce et al. (2021) results significantly underestimated the debris thickness at certain locations near the terminus of the glacier. For instance, at AWS2, the observed debris thickness was approximately 1.13 m, whereas the inverted thickness was only 0.47 m.

The simulation was conducted at a spatial resolution of 300m and a temporal step of 1 day. The primary meteorological drivers, such as precipitation and temperature, were calibrated using data from meteorological stations. We employed statistical methods to downscale all meteorological inputs to a resolution of 300 m (for more details, please refer to the supplementary material). The simulation grid was constrained using the glacier boundaries from Xie et al. (2023), and no ice flow dynamic adjustments for the glacier were considered. In this study, we also conducted a simulation on the debris-free Pasu Glacier situated adjacent to the Batura Glacier to make a comparative study of mass and energy balance. We assumed that Pasu Glacier experiences similar climatic conditions to Batura Glacier. The physical parameters used for this simulation are identical to those from AWS1 on Batura Glacier (see the Section 3.2.2) and we compared the simulated mass balance with the geodetic mass balance to test the extension of these parameters.

### 3.2.3 Parameters calibration/ validation

In this study, we used value ranges for most parameters which have been acquired from empirical equations, large extent observations, or physical process simulations in previous studies e.g., Reid and Brock (2010), Mölg et al. (2012), Hoffman et al. (2016), Zhu et al. (2020), and Sauter et al. (2020). Since the model is very complex, it was necessary to constrain the number of calibrated parameters to limit the modeling effort. Through sensitivity analysis at AWS1, we identified four parameters that have significant impacts on simulating mass balance: ice albedo and roughness length of ice, which constrain ice melting through the radiative and turbulent energy fluxes, respectively; and firn albedo and roughness length of firn, which control the snow evolution processes. By adjusting these parameters within a specific step range, our goal was to achieve the closest match between simulated albedo and longwave radiation and their observed values using a self-defined $\text{RMSE}_{score}$. The $\text{RMSE}_{score}$ is calculated as Eq.7.

$$\text{RMSE}_{score} = \sum_{k=1}^{n} \sqrt{\frac{1}{m}\sum_{i=1}^{m}(obs\_std_{k,i} - sim\_std_{k,i})} \tag{7}$$

Where $n$ represents the number of variables, $obs\_std_k$ and $sim\_std_k$ represent the standardized observed and simulated values of kth variable. The standardization is achieved through min-max normalization. For the purpose of comparison, the final $\text{RMSE}_{score}$ is presented as a standardized result ranging from 0 to 1. A smaller $\text{RMSE}_{score}$

indicates better performance of the model. By comparing the $RMSE_{score}$, we can easily determine the optimal
values for calibrating the parameters (Figure S1). The final determined values for the selected parameters are show
in Table S2. With these parameters, the RMSE between simulations and observations on albedo and outgoing
longwave radiation are 0.09 and 18.93 W/m$^2$, respectively, and there is a high degree of correlation between
observations and simulations on annual variations, with Pearson correlation coefficients (r) of 0.83 for albedo and
0.86 for outgoing longwave radiation (Figure S2). After determining the primary parameters, we fine-tuned some
independent parameters such as albedo timescale, albedo depth scale, temperature threshold of rain/snow ratio,
ensuring a comparable level of simulated mass balance with geodetic mass balance. The simulated mass balance
agrees well with the geodetic mass balance, with an average bias of 0.27 m w.e. In particular, there is a strong
agreement between the results from Hugonnet et al. (2021) and our simulations in terms of the trend observed from
2000 to 2020 (Figure 3). This indicates that the parameters used in our study can reliably estimate the mass and
energy budget.
A point simulation at AWS2 was conducted to calibrate and validate the parameters required to simulate energy
balance in debris layers. Following Giese et al. (2020), we evaluated the model parameters by optimizing the
agreement between the simulated surface temperature and the surface temperature recorded by AWS 2 (the
temperature probe is buried ~ 2 centimeters below the debris surface). The parameters calibrated at AWS1 were
applied unchanged to AWS2, with adjustments only made to the debris thermal conductivity and debris albedo
during the simulation process. The calibration process can be observed in Figure S3. Figure 4 depicts the
comparative analysis of the observed station temperature and the simulated temperature, using the optimized values
for debris thermal conductivity and albedo, revealing a strong consistency between the two over time, with a
correlation coefficient of 0.87, although there is a tendency to underestimate the temperature in late summer and
autumn, and overestimate temperature in late winter. The correlation of observed and simulated temperature for the
annual cycle is 0.96, while the RMSE during the simulation period is 0.86 ℃.
The parameter evaluation process at AWS 2 supports the applicability and scalability of the parameters
calibrated at AWS1 to other parts of the glacier. Based on the final parameters determined (Tables S2 and S3), the
simulated mass balance for the entire glacier is estimated to be -0.23 m w.e. yr$^{-1}$ (2000-2016). This value closely
aligns with the geodetic mass balances derived from remote sensing (-0.18 m w.e.yr$^{-1}$, spanning the years 2000-
2016, Brun et al. (2017); -0.39 m w.e.yr$^{-1}$, covering the years 2000-2009, Bolch et al. (2017); and -0.24 m w.e.yr$^{-1}$,
covering the years 2000-2014, Wu et al. (2020)). This further supports the robustness of parameter transfer across
the glacer.

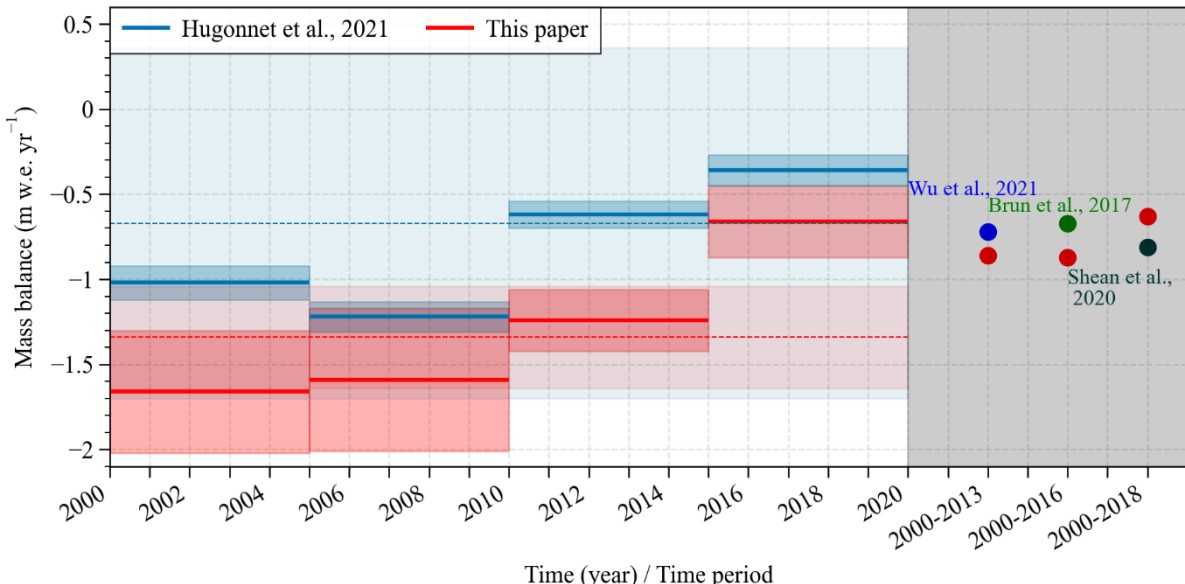


**Figure 3** Comparison of simulated and geodetic mass balance over different time periods. To assess the performance
of our model, we compared the simulated mass balance with estimates derived from geodetic observations. However,
it is important to acknowledge that this approach introduces a degree of dependence between the two results since
some model parameters were calibrated using the geodetic mass balance.

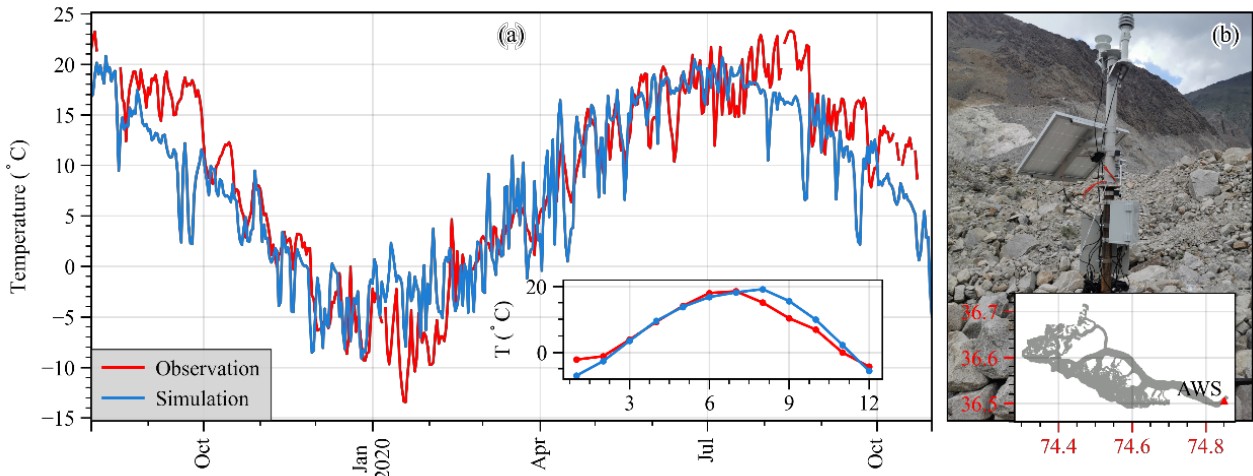


**Figure 4** (a) Observed and simulated surface temperature at AWS 2. (b) Photograph and location of AWS 2 on
Batura Glacier. AWS 2 collects data at both daily and hourly intervals, this study utilizes daily records for
analysis.

4 Results and discussions
4.1 Glacier climatic-mass-balance dynamics and corresponding energy budgets
4.1.1 Energy budgets
During 2000-2021, the surface net radiation of the Batura Glacier accounted for the largest proportion of total
energy heat flux (46%), followed by sensible heat flux (23%). Latent heat flux (-18%) and conductive heat flux
(17%) demonstrated a similar magnitude of contribution to the total energy heat flux, albeit with opposite sign
(Table 1).
The net shortwave radiation accounted for 85% of the total energy influx (77 W/m$^2$), while sensible heat
constituted 15% (14 W/m$^2$). Regarding energy sink components, net longwave radiation contributed 57% (52 W/m$^2$),
melt heat 20% (18 W/m$^2$), latent heat 12% (11 W/m$^2$), and conductive heat 11% (10 W/m$^2$). In terms of the energy
components that contribute to glacial mass loss, sublimation latent heat accounted for approximately 38%, while
the energy for snow/ice melting constituted 62%. For the Batura Glacier, roughly 32% (29 W/m$^2$ out of 91 W/m$^2$)
of the surface energy influx was consumed by glacier mass loss, a proportion similar to that of Muztag Ata No.15
Glacier, which is also situated in the Westerly influenced area (30%, 26 W/m$^2$ out of 89 W/m$^2$) (Zhu et al., 2017).
However, it is worth noting that the melting heat of the Batura Glacier was significantly higher than that of Muztag
Ata No.15 Glacier (~2 W/m$^2$), possibly due to differences in surface debris cover between the two glaciers.
During the period of accumulation, a notable proportion of 73% of the energy influx of the Batura Glacier was
expended through net longwave radiation, with 15% of the energy utilized for snow/ice sublimation, leaving the
remaining portion dedicated to thermal conduction within the debris cover or snow layer. In contrast, throughout
the ablation season, the energy influx was mostly from net shortwave radiation, specifically amounting to 133 W/m$^2$.
The conductive heat flux exhibited by the Batura Glacier diverged significantly from debris-free glaciers, such as
the Guliya ice cap (Li et al., 2019). In the Batura Glacier, a considerable portion of the energy influx at lower
elevations was absorbed by the debris cover, resulting in higher surface temperatures compared to the lower layers,
thus yielding heat transfer towards the debris-ice interface. Conversely, in the accumulation area, the primary source
of energy was dedicated to heating the snow layer. It became evident that during the ablation season, the debris
cover assumed a more prominent role, ultimately leading to an overall negative thermal conduction.




**Table 1** The energy budget on Batura Glacier. $lw_{in}$ and $lw_{out}$ denote incoming and outgoing longwave radiation, $sw_{in}$ and $sw_{out}$ denote incoming and outgoing shortwave radiation, $sh$ and $lh$ represent the sensible heat flux and latent heat flux, $g$ represents conductive heat flux, and $me$ represents melt energy. All values are expressed in W/m².

| Periods | $lw_{in}$ | $lw_{out}$ | $sw_{in}$ | $sw_{out}$ | Net lw | Net sw | Net radiation | | sh | | lh | | g | | $me$ |
|---|---|---|---|---|---|---|---|---|---|---|---|---|---|---|---|
| | | | | | | | — | % | — | % | — | % | — | % | |
| Annual average | 212 | -264 | 249 | -172 | -52 | 77 | 25 | 42 | 14 | 23 | -11 | 18 | -10 | 17 | 18 |
| Ablation (6-9) | 231 | -293 | 345 | -212 | -62 | 133 | 71 | 65 | -7 | 6 | -15 | 14 | -16 | 15 | 33 |
| Accumulation (10-5) | 202 | -249 | 187 | -153 | -48 | 34 | -12 | 19 | 32 | 52 | -10 | 16 | -8 | 13 | 0 |

4.1.2 Mass balance history

The results from the EMB model show that the average mass balance of the Batura Glacier during the studied period was -0.262 ± 0.561 m w.e. yr⁻¹ (Table 2). The glacier experienced its highest positive mass balance in 2010 (0.32 m w.e. yr⁻¹) and its greatest negative mass balance in 2001 (-1.19 m w.e. yr⁻¹). Snowfall was the primary source of glacier mass gain, accounting for 89% of the total mass gain. Refreezing mitigated the internal melting caused by radiation penetration and contributed to 11% of the mass accumulation. Glacier melting constituted 92% of the mass loss, while sublimation/evaporation, which exhibited minimal interannual variability, contributed only 8% to the mass loss.

The model simulations show a decline in glacier ablation after 2008, accompanied by a decrease in the absolute magnitude of the mass budget over the study period (Figure 5a). Independent measurements of thinning rates at the glacier terminus measured by ground-penetrating radar, declined from 4.58 m yr⁻¹ between 1974-2000 to 0.59 m yr⁻¹ after 2000 (Gao et al., 2020), implying a similar reducing trend in surface melt rate, which further supports the EMB results. The striking decrease in thinning rates at Batura Glacier for the periods 1974-2000 and 2000-2017, and decline in modeled ablation since 2008 might be linked to regional climate fluctuations. Previous studies based on station observations have indicated a notable cooling trend in the upper Indus River basin during the summer months, particularly in July, September, and October, from 1995 to 2012 (Hasson et al., 2017). Moreover, there was a lack of long-term warming during the winter months over the same period (Hasson et al., 2017). Forsythe et al.

(2017) suggested that the summer temperature in the Karakoram was relatively low and exhibited a decreasing trend
due to the influence of the Karakoram vortex (KV). This influence may have contributed to the notably higher
positive accumulated temperatures pattern observed from 1970 to 2000 compared to those recorded after 2000, as
shown in Figure 4b of Forsythe et al. (2017). Our analysis on air temperature in the Hunza basin from 1980~2020,
utilizing ERA5 data, corroborates these findings (Figure S4).
As shown in Figure 5b, the variations in internal mass balance and surface mass balance are generally
consistent throughout the year, both showing a negative mass balance from June to September. During this period,
there was a high shortwave radiation and, consequently, a great amount of shortwave radiation penetrated into
snow/ice. This increased ablation resulted from penetration radiation, coupled with relatively high temperature,
reducing the rate of refreezing, and thus causing a negative internal mass balance. The mass budgets in May and
October were transitional between accumulation and ablation periods. The seasonal pattern on mass balance
observed in this study is generally similar to that of the Siachen Glacier, East Karakoram presented by Arndt and
Schneider (2023). Both glaciers exhibit a characteristic of winter/spring accumulation. However, the modeled
meltwater during the ablation season was found to be significantly lower for Siachen Glacier compared to Batura
Glacier. It is worth noting that Arndt and Schneider (2023) did not consider the impact of supraglacial debris cover
on glacier melt, which is known to be substantial (Agarwal et al., 2016). Even without considering the debris cover,
the mass balance of Siachen Glacier, as indicated by Arndt and Schneider (2023), can still remain in equilibrium,
largely depending   on the precipitation and temperature driving data. On the other hand, in the simulation study
conducted by Kumar et al. (2020), Siachen Glacier exhibited a negative mass balance during the same period, with
the average temperature and precipitation being higher than those used by Arndt and Schneider (2023). This suggests
that simulation results can be considerably influenced by model inputs, and this will be discussed in Section 4.5.

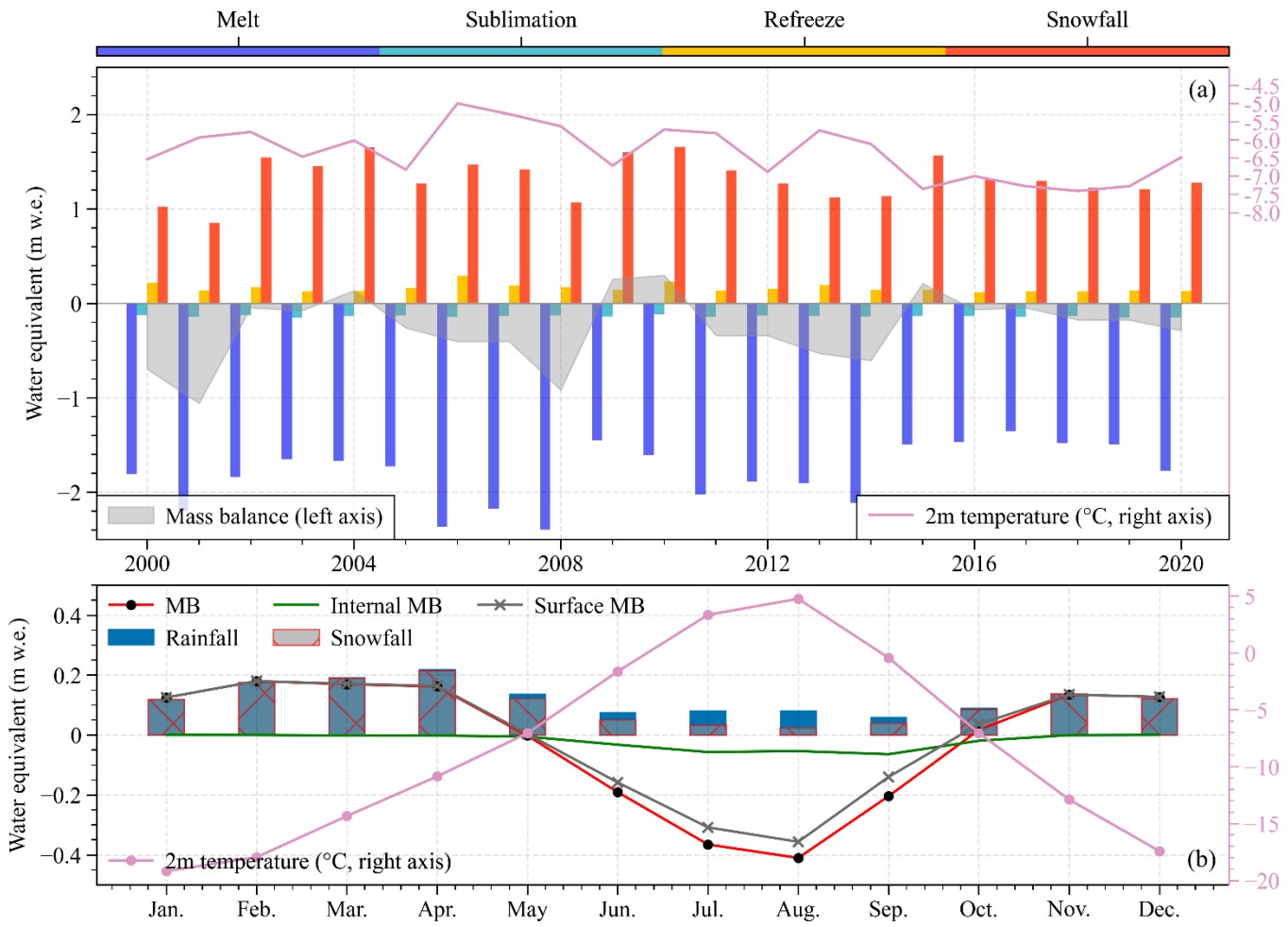


**Figure 5** Interannual (upper panel) and mean monthly (lower panel) characteristics of the glacier-wide average of

mass components on Batura Glacier over the study period. The MB denotes mass balance. The 2m temperature is

obtained from the simulated results.

**Table 2** Mean values of the mass balance components of Batura glacier over 2000 to 2020.

|  | Mass balance | Snow accumulation | Surface melt | Refreezing | Sublimation |
|---|---|---|---|---|---|
| Values (m w.e. yr⁻¹) | -0.262±0.561 | 1.325±0.174 | 1.613±0.394 | 0.162±0.125 | 0.136±0.005 |
| Proportion of mass gain (loss) (%) | — | 89 | (92) | 11 | (8) |


Over the study period, the glacier demonstrated a positive rate of annual mass balance change of 0.023 m w.e.

yr⁻², indicating the glacier's mass balance was becoming less negative and approaching equilibrium between 2000-

2020 (Figure 6a, b and d). Particularly noteworthy is the trend of decreasing mass loss across the ablation zone,
which is particularly pronounced in the junction where debris cover and bare ice intersect and the tributary where
debris cover is thin or absent (Refer to debris cover in Figure 6e), which indicates a reduction in melt (Figure 6b).
Given the rate of mass balance change over time (reduction of melt) is highest in these areas, the mass changes in
these areas probably have a large impact on the trend of decreasing negative mass balance.

436        Across the entire accumulation zone, a slight decrease in mass gain over the 2000-2020 period was observed,

with a more pronounced reduction in mass gain observed on the southern flank of the accumulation area, likely
associated to diminished winter snowfall. From a mass budget perspective, the glacier's mass balance appears to be
approaching equilibrium, likely due to the reduced melting during the months of June and July (Figure 6c). For
instance, in years characterized by a positive mass balance, such as 2010, the duration of mass accumulation in
spring extended, accompanied by minimal mass loss during June and July. The glacier's mass balance generally
followed a cyclic pattern spanning roughly five-seven years. The mass balance has become more negative after
2016, possibly indicating a phase of reduced snow accumulation gain (Figure 6c).

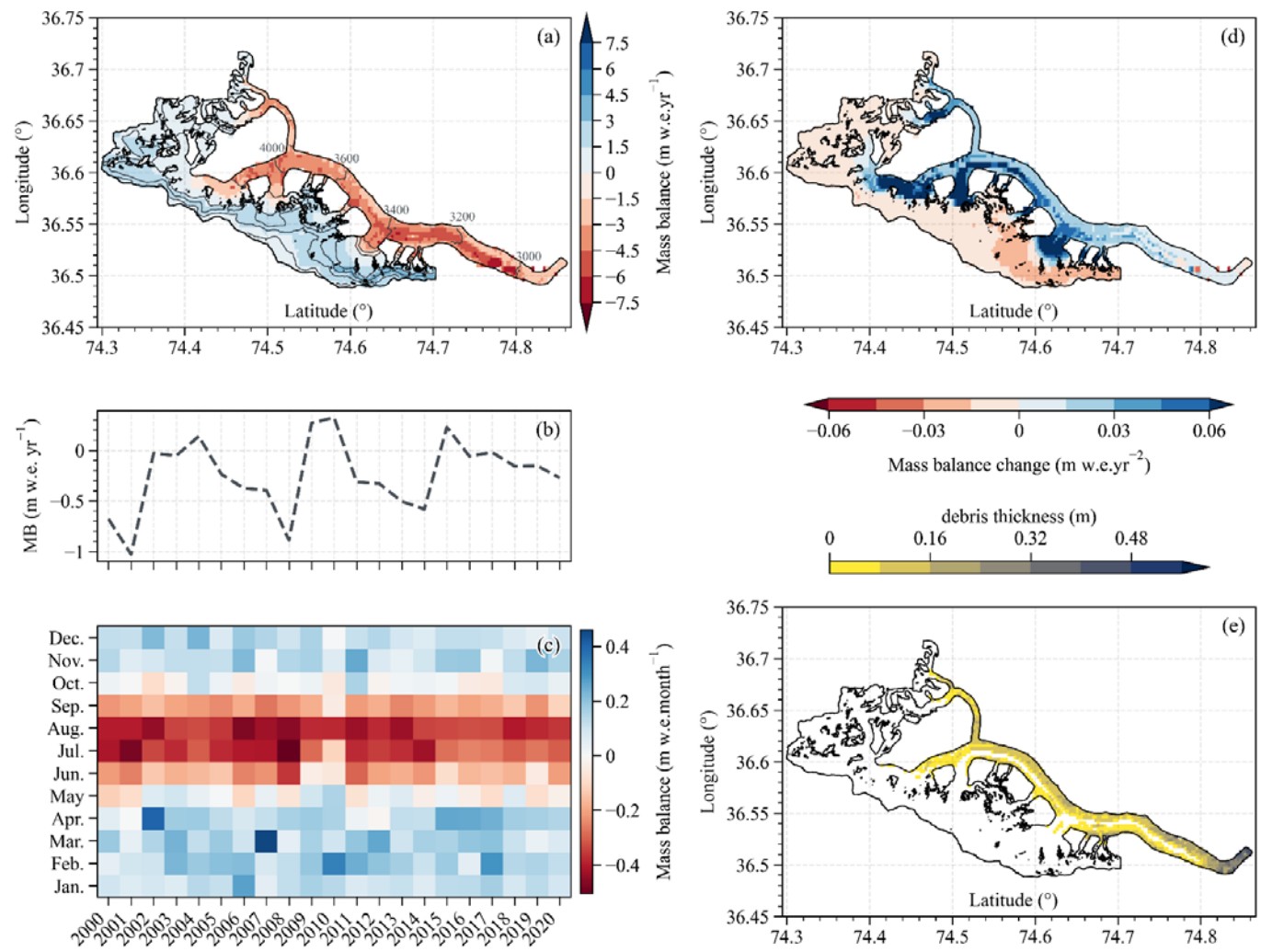


**Figure 6** Spatial distribution of the annual mass balance over the 2000-2020 period (a). Time series of modeled

annual (b) and monthly (c) mass balance from 2000-2020. Spatial distribution of the annual mass balance change

rate over the 2000-2020 period (d). Spatial distribution of debris thickness (e)

4.2 Energy and mass budgets along the altitudinal profile

A significant heterogeneity of mass balance was observed in the Batura Glacier. The mass gain in the glacier

accumulation zone can reach up to almost 2 m w.e., whereas terminus melting exceeded 4 m w.e. between 3000-

3800 m, with the maximum melting of 4.6 m w.e. occurring within the elevation range of 3350-3450 m. Mass

balance exhibited discernible altitude-dependent distribution, whereby the most substantial melting was observed

not at the terminus but rather in the range between 3000 and 3400 m (Figure S5a).

A comparative analysis was performed to understand the variations in mass balance across different elevation

zones between Batura Glacier and Pasu Glacier. The equilibrium line altitude (ELA) of the Batura Glacier (4500 m)

was significantly higher than that of the Pasu glacier (4150 m). Below the ELA, both glaciers exhibit gentle overall
slopes, leading to high receipt of solar shortwave radiation. As shown in Figure 7, the net radiation of the Batura
Glacier was significantly larger than that of the Pasu glacier, primarily attributable to surface albedo disparity. The
Pasu Glacier's surface primarily comprises firn or ice, whereas the Batura Glacier is largely covered with fragmented
rocks with associated lower albedo. Evidently, the melt energy for the Batura Glacier is less than that of the Pasu
Glacier, chiefly due to heat conduction between debris layers, which absorb a substantial amount of energy. Overall,
the Batura Glacier demonstrated an "arch-shaped" melt energy pattern from its terminus to the ELA, in sharp
contrast to the "slope-increasing" pattern exhibited by the Pasu Glacier. This altitude-dependent spatial energy
distribution pattern also affects that of the glaciers' melt (Figure S5).
Within the regions spanning from the ELA to the zones of maximum snow accumulation (Batura: 4500-5400
m, Pasu: 4150-5400 m), glacier mass accumulated rapidly due to significantly heavy snowfall (Figure S5). Turbulent
heat exchange intensifies within this altitude range, with melt energy approaching zero. A modest amount of melting
resulted in mass accumulation within the snowpack through refreezing (Figure S5). At altitudes exceeding 5200 m,
net radiation, turbulent exchange, and conductive heat flux did not demonstrate significant variations. Net radiation
was dominated by longwave radiation, and the snow's surface temperature surpassed the air temperature. The glacier
acted as an energy source, transferring energy to the atmosphere to maintain energy balance. While the maximum
snowfall on the Batura Glacier was similar to that on the Pasu Glacier, the accumulating area was larger. For instance,
in the region above 7000 m, up to 1 m w.e. of snowfall was observed on the Batura Glacier (Figures S5). Changes
in precipitation not only induced net radiation variations due to snow albedo feedback but also triggered outgoing
longwave radiation and sensible heat variations through alterations in surface temperature. This trait aligned with
some of the other glaciers in this area, as well as some glaciers in the West Kunlun and Pamir (Li et al., 2019; Zhu
et al., 2017; Bonekamp et al., 2019). However, the Batura Glacier exhibited more negative mass balance compared
to these glaciers including the Pasu glacier (The geodetic mass balance, as reported by Brun et al. (2017), is -0.01
$\pm$ 0.05 w.e.m yr$^{-1}$, while the simulated mass balance in this study is 0.01 $\pm$ 0.26 w.e.m yr$^{-1}$, both for the period from
2000 to 2016.).

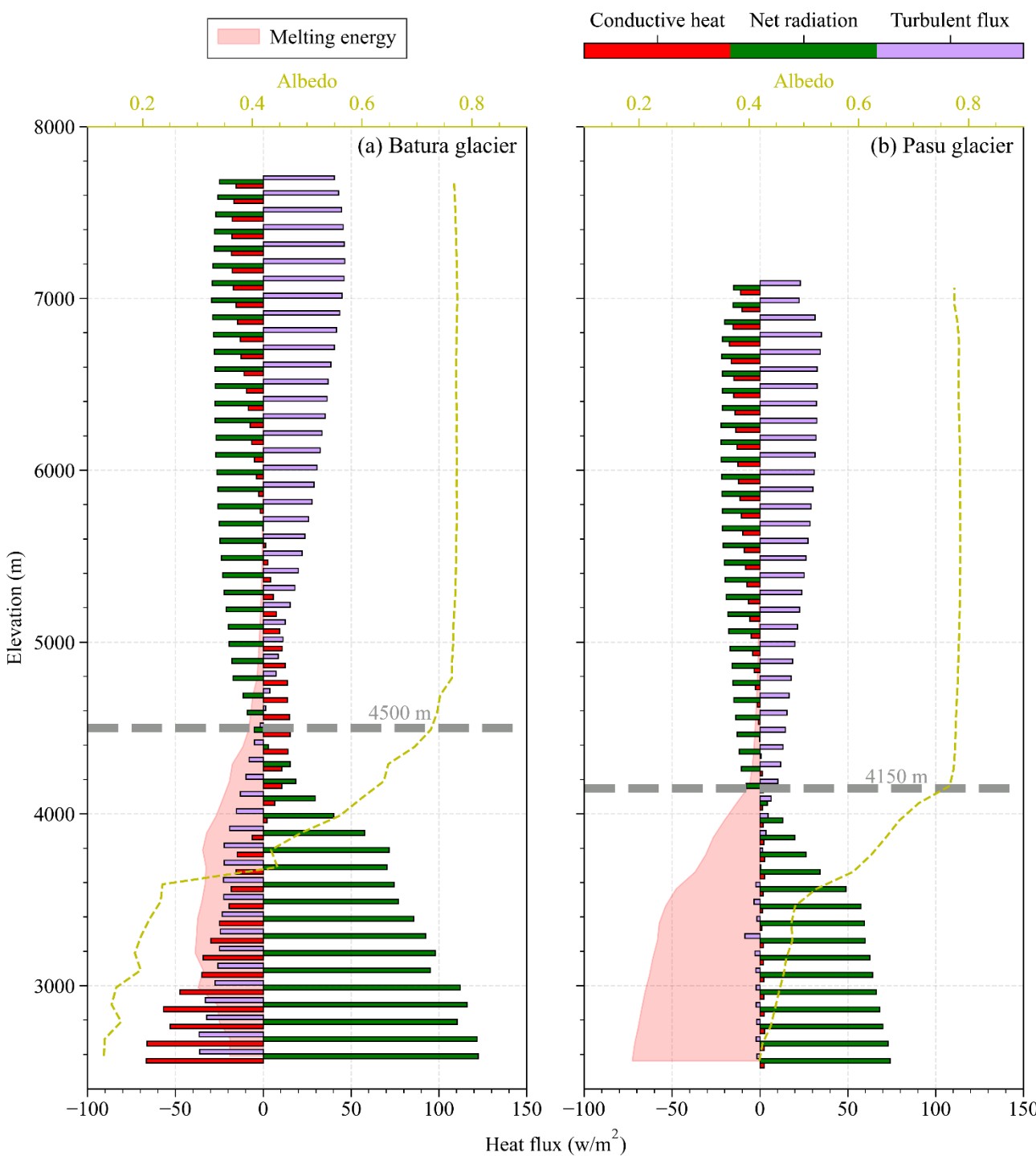

**Figure 7** Altitudinal distribution of the primary energy balance components for (a) Batura Glacier and (b) Pasu glacier.

4.3 Impact of debris cover on glacier mass balance

Our findings revealed that the presence of supraglacial debris led to a notable 45% reduction in negative mass

balance of the Batura Glacier. Specifically, in the absence of debris, the mass balance exhibited a value of -0.48 m
w.e. yr[-1], whereas with the inclusion of debris, this value decreased to -0.26 m w.e. yr[-1], likely due to the insulating
effect of debris on melt rate In contrast, a similar modeling experiment conducted in the Karakoram found that the
Baltoro Glacier experienced a reduction in ablation by approximately 35% when debris was excluded (Groos et al.,
2017). Moreover, glaciers in the Central Karakoram National Park, Pakistan, showed a 24% decrease in modeled
ablation when debris was excluded (Minora et al., 2015). It's important to note that these contrasting findings with
respect to the impact of debris cover on glacier mass balance in the Karakoram can be attributed to differences in
the models employed, their configurations, and the thickness distribution of debris cover. The latter directly impacts
the spatial characteristics of sub-debris melting intensity (Compagno et al., 2022).
On a daily or monthly basis, the impact of supraglacial debris on the Batura Glacier manifested most
prominently during the ablation season, as depicted in Figure 8a and b. On an interannual scale, supraglacial debris
had a significant impact on mass balance of the Batura Glacier; however, it did not induce alterations in its overall
temporal fluctuations or trends (Figure 8c). This was mainly because the simulation process did not include the
influence of changes in the debris cover distribution over time on mass balance.
The debris had a significant protective effect, effectively mitigating glacier ablation. This effect was most
pronounced in August, a period characterized by high air temperatures. During May and June an extensive snow
cover blanketed the Batura Glacier. When supraglacial debris is included in energy balance processes, the snow
layer absorbed a greater amount of heat from the atmosphere through thermal conduction, thereby leading to
accelerated melting. As the snow progressively melted and the debris became exposed, the surface albedo
experienced a rapid decline spanning from July to October. This transition resulted in the debris absorbing a greater
portion of incoming shortwave radiation, much of which is returned to the atmosphere as emitted longwave radiation
or sensible heat, consequently yielding a reduction in the melting energy available (Figure 8b). Statistical analysis
revealed that when supraglacial debris was not considered, the average net radiation decreased by 14 W/m[2]. The
most substantial reduction was observed in May, with a reduction of approximately 20 W/m[2].


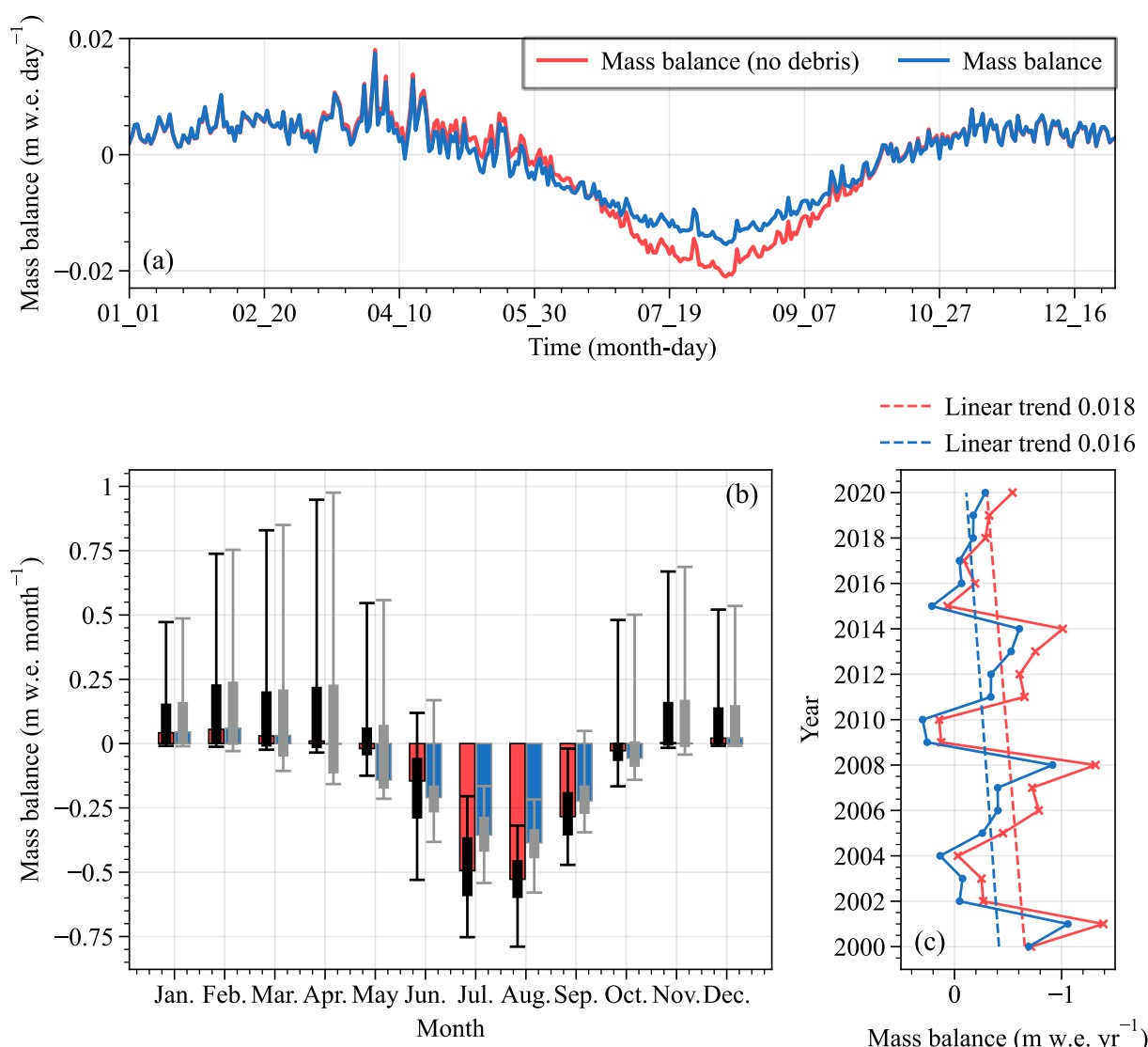

**Figure 8** The difference between modeled mass balance with (blue lines and bars) and without debris cover (red lines and bars): (a) daily mass balance; (b) monthly mass balance; and (c) annual mass balance trend.

4.4 The energy controls of sub-debris melt

We conducted additional investigations to understand how the supraglacial debris affect the ice ablation. In the case of the Batura Glacier, the presence of supraglacial debris reduces the average albedo of the glacier, thereby increasing net shortwave radiation. Notwithstanding the observed augmentation in net radiation, an attenuation in melt was recorded. To investigate the impact of debris on energy-driven melting, this study conducted a statistical analysis of the energy balance for scenarios with and without debris coverage in the specific area characterized by the presence of debris (Figure 9). The results indicated that while the presence of debris did amplify the net radiation income, the available energy for melting is reduced by the sum of longwave radiation emission, sensible heat, and

thermal conduction within the debris (an average decrease of 25 W/m$^2$).

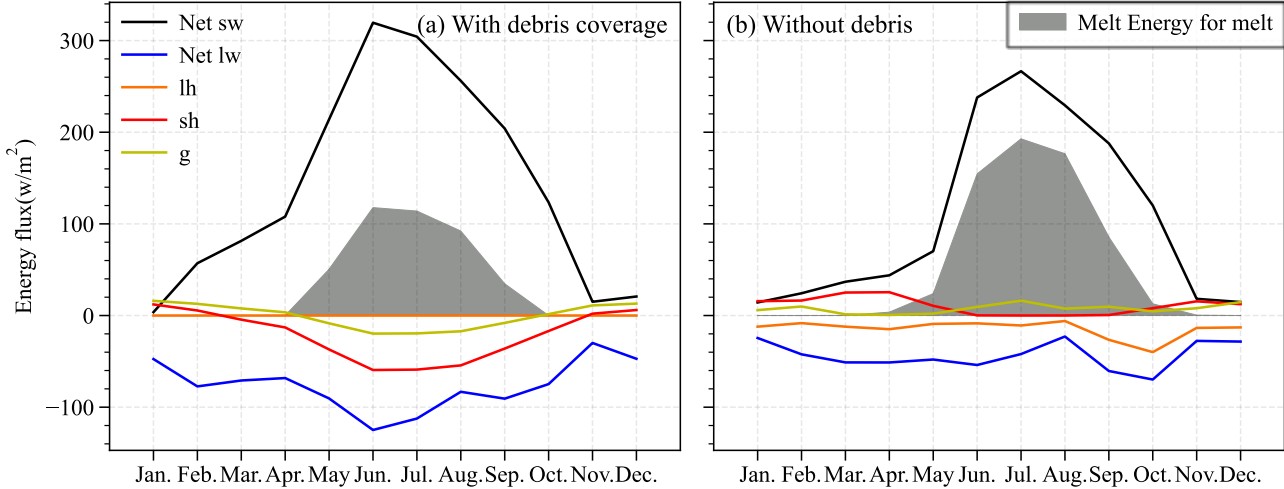


**Figure 9** Annual cycles of energy budget components (a) with and (b) without debris coverage for the currently
debris-covered area on Batura Glacier.
During the ablation season (June to September), when accounting for the presence of debris, the glacier's
energy income, represented by net shortwave radiation, witnessed an augmentation of 61 W/m$^2$. Meanwhile, the
energy output increased by 116 W/m$^2$, comprising net longwave radiation (50 W/m$^2$), sensible heat (42 W/m$^2$), and
conductive heat (24 W/m$^2$). Consequently, this led to a reduction of 45 W/m$^2$ in latent heat of melt (sublimation heat
of the debris layer, which was not considered when deducting the 11 W/m$^2$ for sublimation heat without debris cover)
(Figure 9). In light of these observations, it can be concluded that the influence of debris cover on glacier melt is
twofold. Firstly, it reverses the net direction of turbulent heat fluxes at the glacier surface. Secondly, it alters the
heat flux reaching the glacier through thermal conduction. The former aspect primarily emanates from the heating
of the debris layer due to shortwave radiation, causing the debris temperature to surpass the atmospheric temperature.
Consequently, the glacier transfers heat to the atmosphere, effectively acting as an energy source. This finding aligns
with earlier research results, as exemplified by Steiner et al. (2018) and Nicholson and Stiperski (2020). Regarding
the second aspect, we conducted an analysis that considered the thermal conduction occurring within both the debris
and ice layer, as well as the energy equilibrium within each layer. When the heat gained from net radiation was
conducted within the debris layers (the radiation penetration of the debris was neglected), it could be consumed to
heat the debris, thereby satisfying the energy balance within and between the debris layers.
At the interface between debris and ice, heat exchange exhibits pronounced seasonal variations, with notable
altitudinal gradients, particularly during the accumulation period (Figure 10). In the ablation season, a debris layer
is very quickly warmed by solar radiation before cooling back close to zero after sunset. The temperature of surface
debris rises, transferring heat into the interior of the debris (Reid et al., 2012). However, the energy reaching the
debris-ice interface is predominantly influenced by the thickness of the debris layer. Below 2900 m, where the debris
thickness exceeds 20 cm, the energy at the debris-ice interface is less than 90 W/m². At altitudes above 3200 m
where the debris thickness is less than 11 cm, the energy at the debris-ice interface increases to 140 W/m² (Figure
10). At these altitudes the debris thickness remains relatively constant, and correspondingly, the energy flux at the
debris-ice interface exhibits minor fluctuations. Despite Collier et al. (2015)'s suggestion that near-surface air
temperature is generally a stronger driver of melt rates below debris, our findings from the energy at the debris-ice
interface, in conjunction with Figure S6, imply that this relationship may not hold true during the ablation season
in high-altitude regions. During the accumulation season, the energy at the debris-ice interface is negative, with the
glacier transferring heat to the debris layer. This significantly affects the upwelling longwave radiation and sensible
heat flux at the debris surface. Thinner debris layers result in more heat transfer from the glacier to the debris (Figure
10b). In contrast to the ablation period, the energy at the debris-ice interface steadily increases with altitude during
the accumulation season. This difference may be attributed to snowfall causing substantial variations in the surface
energy balance process during the accumulation period compared to the ablation season. Overall, altitudes below
2900 m are identified as the less sensitive zone for Batura Glacier's ablation. Conversely, the areas where debris
cover and bare ice intersect emerge as highly sensitive zones for melting, with the average thickness of debris in
these regions being less than 2.3 cm.

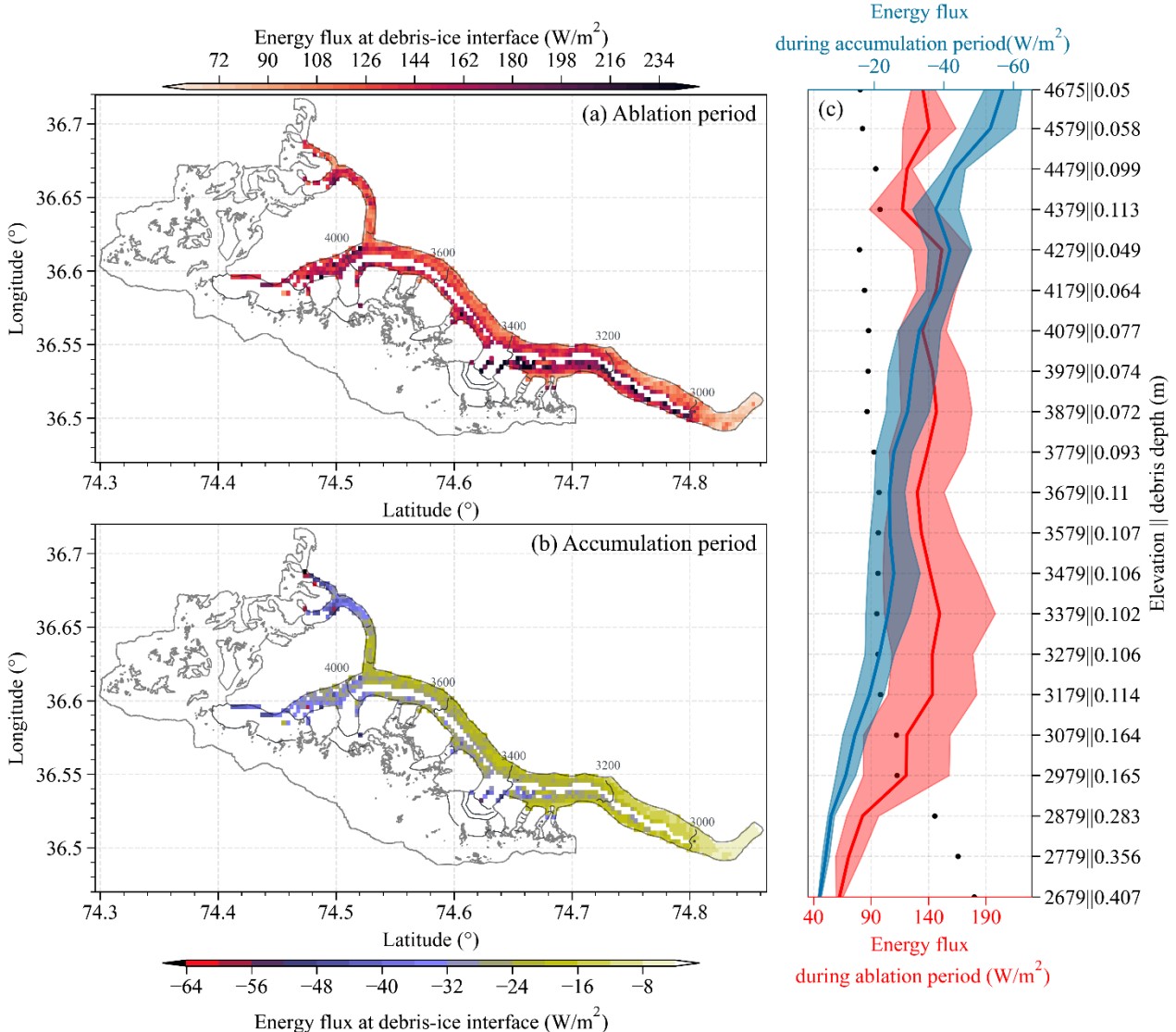

**Figure 10** Spatial distribution of the mean energy flux at the debris-ice interface during ablation (a) and

accumulation (b) periods. An elevation-dependent distribution of the debris-ice energy flux in each season is

shown in (c).

The process of heat conduction within the debris was clearly illustrated in our study through an analysis of

temperature changes within debris of varying thicknesses (Figure 11). During the ablation season, for thinner debris

(Figure 11b, location P1), achieving a stable ice surface at zero °C necessitates a temperature difference of 2.5°C

within the uppermost 0.015 m (comprising 3 layers), with an average temperature decrease of 1.7°C per 0.01 m

increment. Conversely, in the case of thicker debris (Figure 11f), with a depth of 0.2 m (20 layers), the temperature

alteration amounts to 8°C, accompanied by a vertical temperature gradient of 0.4°C per 0.01 m increment.

Consequently, with respect to the upper layers, thin debris is more likely to conduct a greater amount of heat. At the

interface between the surface ice and overlying supraglacial debris, the temperature difference at P1 (0.035-0.045

m) was 2.5 °C with a vertical gradient of 2.5 °C/0.01m. At P5 (0.42-0.55 m), the vertical gradient of temperature

was 0.61 °C/0.01m (Figure 11). This indicates that in areas covered by thin supraglacial debris, more energy was

transferred from the debris to the glacier, resulting in a greater amount of latent heat being released by the glacier.

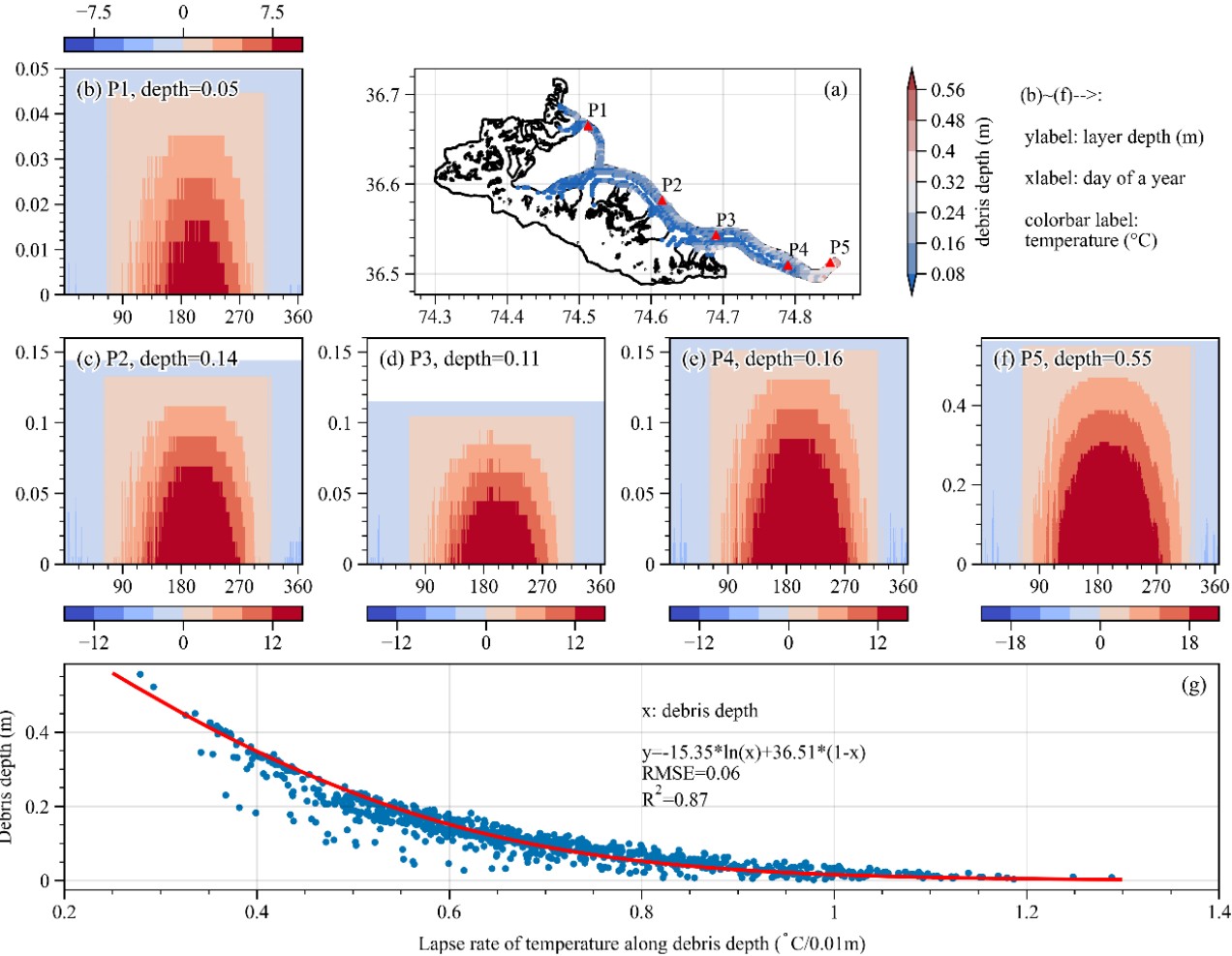

**Figure 11** Temporal variations of debris temperature across different depths throughout a year. Temperature

profiles at specific points in (a) are displayed in (b)~(f). The relationship between temperature lapse rate and

debris depth is presented in (g).

When the thickness of the debris is comparable, the vertical temperature gradient within the debris exhibits a

corresponding similarity (P2, P4), except for slight deviations primarily observed at the surface. These variations

are primarily attributed to discrepancies in both air temperature and surface temperature of the debris between the

two points. Throughout the accumulation period, net shortwave radiation remained limited, leading to low

temperatures and causing the debris temperature to either reach or drop below freezing point. As a result, the rate

of heat conduction process decelerated, thereby mitigating the influence of the debris on glacier melting.

To quantify the relationship between the thickness (x) of the debris layer and the vertical temperature gradient

(y), we computed the average temperature gradient for individual pixels within the debris-covered area during the
ablation period and conducted regression analysis (Figure 10g). According to Eq. 8, an increase in debris layer
thickness corresponds to a reduction in the vertical temperature gradient. Combined with Eq. 4 & 5, the heat
conduction to the interface between the debris layer and the glacier will also decrease, leading to diminished
availability of latent heat for glacier melting. As the thickness of the debris layer approaches minimal values, the
heat originating from a temperature difference of approximately 20°C is used for melting. This fundamentally
quantifies the impact of debris cover thickness on melt and further explains the differences in mass balance shown
in Figure S5.
$$y = -15.35\ln(x) + 36.5(1 - x) \tag{8}$$


4.5 The potential uncertainties and limitations

602       The parameter settings significantly influence simulation results. Of the six calibration parameters, the

simulation results are highly sensitive to firn albedo, ice roughness length, and debris albedo (Figure S1 and S3).
The largest changes are observed when varying the debris albedo. When the debris albedo decreases to 0.1
(approximately a 23% change in albedo from the calibrated value), the melt increases by about 3.4%. With a 100%
increase in debris albedo (0.26), the melt decreases by approximately 14%. This magnitude of sensitivity is
consistent with the findings of Giese et al. (2020) on Changri Nup Glacier in the Himalayas. The calibrated
parameters ice and firn roughness length lie on the margin of the range, implying that a larger range may be
beneficial or that a parameter not considered in calibration is not chosen optimally. However, extending the limits
of these parameters would result in physically unrealistic values. Due to the complexity of the model, we did not
calibrate all parameters. Instead, we identified the aforementioned six parameters through sensitivity analysis.
Besides the calibrated parameters, certain factors, such as the rain and snow separation threshold influence the
simulated mass balance. In this study, we constrained these parameters using geodetic mass balance.

614       Apart from the model-inherent parameters, the model's input dataset presents considerable challenges during

calibration and introduces uncertainty into the results (Arndt and Schneider, 2023). While HAR data has been
applied in glacier mass balance simulation studies (e.g., Huintjes et al. (2015b) and Groos et al. (2017)), its
applicability in the Karakoram mountains remains uncertain (Groos et al., 2017) due to the majority of ground
validation being conducted on the Tibetan Plateau (Maussion et al., 2014). Additionally, uncertainties can also be
introduced by the calibration methods and downscaling schemes of the climatic factors, as evident from the

comparison of our study with results from Groos et al. (2017). Initially, Groos et al. (2017) downscaled HAR Version 1 data to 30 m resolution using interpolation for glacier mass balance simulations in the Karakoram. In this study, we first calibrated temperature and precipitation in HAR Version 2 using station observations and then employed statistical downscaling to achieve a 300m resolution for energy balance research, incorporating radiative downscaling that accounts for complex topography. While both results of Groos et al. (2017) and this study compare well with station observations, discrepancies exist in temperature and precipitation on Batura Glacier. For example, Groos et al. (2017) reported a temperature of 5.0 °C during the ablation season at ~4,060 m a.s.l., while this study recorded 1.7°C at the same elevation. Annual precipitation for Batura Glacier is ~960 mm in this study compared to 1059 mm in Groos et al. (2017). These differences resulted in significant spatial disparities between the two simulated results (Figure 5a of this study and Figure 6 of Groos et al. (2017)). Although the multi-year average mass balance in this study aligns more closely with geodetic mass balance compared with that of Groos et al. (2017), it remains challenging to determine which result can better capture the spatial characteristics of glacier mass balance due to a lack of knowledge about meteorological conditions in high-altitude glacierized regions and insufficient characterization of surface features like ice cliffs and supraglacial ponds in both models. Therefore, as highlighted by Collier et al. (2013), this uncertainty can only be minimized through additional high-altitude observations and more reliable downscaling approaches, such as dynamic downscaling.

The spatial resolution of a glacier model can impact simulation results, particularly in debris-covered areas. To investigate this effect, we conducted comparative simulations with varying resolutions on a small section of the Batura Glacier terminus. We used the 300 m resolution simulation from this study as the benchmark. When increasing the resolution to 100 m (matching the debris data resolution), the average debris thickness showed a minimal difference of 0.01 m compared to the 300 m resolution thickness. However, the spatial distribution of debris thickness exhibited significant discrepancies, especially at the glacier margins (Figure S7a, b). Notably, subsurface melt rate decreased by 2.2% compared to the benchmark (Figure S7e). Since debris albedo was set as a constant value, net radiation remained relatively unchanged. However, the surface temperature decreased by 0.17°C (Figure S7f), accompanied by a 1.9% reduction in sensible heat flux (Figure S7i) and a 2.7% decrease in conductive heat transfer within the debris layer (Figure S7j). These findings demonstrate that while spatial resolution influences the energy fluxes and ablation of debris-covered glaciers, its primary impact lies in the spatial distribution (Figure S7c, d) with minimal effect on average values. This spatial variation primarily stems from the differences in debris thickness captured at varying resolutions. Given the limitations of the employed debris thickness data (Rounce et

al., 2021), we cannot definitively conclude if higher resolution simulations yield results closer to reality. Additionally, the computational cost of high-resolution simulations is substantial. Therefore, this study utilized a coarser grid to capture the overall energy and mass balance characteristics of the glacier. However, the potential for more realistic outcomes with reliable high-resolution debris thickness data is undeniable.

The main limitation of the model lies in the absence of parameterization for the impact of glacier surface features on melting, such as ice cliffs and supraglacial ponds. This omission may lead to an underestimation of the ice melt rate across debris-covered areas, as observed amplifying effects of supraglacial lakes and ice cliffs on glacial melt (e.g., Tedesco et al. (2012), Miles et al. (2016), and Buri et al. (2021)) are not considered. Supraglacial ponds and lakes efficiently transfer heat into glacier ice due to their low surface albedo and active convection. Simulations by Miles et al. (2018) indicated that ponds may contribute to 1/8 of total ice loss in the Langtang Valley, Nepal. Modeling by Huo et al. (2021a) also suggested a substantial increase in ice loss on the Baltoro Glacier in the Karakoram due to the intervention of supraglacial ponds. Supraglacial ice cliffs influence glacier ice melt by creating a direct ice-atmosphere interface with low albedo and exposure to high emissions of longwave radiation from surrounding debris-covered surfaces (Buri et al., 2016). According to Buri et al. (2021), neglecting ice cliffs in Langtang Valley would result in a mass loss underestimation of 17% ± 4% for debris-covered glacier tongues. In most glaciers, interactions generally exist between ice cliffs and ponds/lakes (Buri et al., 2021; Huo et al., 2021a). Therefore, future research should incorporate parameterization for these elements to better understand their impact on glacier melting. However, in the absence of sufficient observations, a limited representation of ponds and ice cliffs in the parameterization of model can introduce additional uncertainty in glacier-wide energy fluxes (Miles et al., 2016).

5 Conclusions and outlook

This study presented a comprehensive investigation into the relationships between supraglacial debris cover, energy fluxes, and mass balance dynamics on the Batura Glacier in the Karakoram. Through simulation analysis, we propose that the presence of debris on the glacier surface effectively reduces the amount of latent heat available for ablation by absorbing solar radiation and preventing it from reaching the ice surface, which creates a favorable condition for the Batura Glacier's relatively low negative mass balance. Furthermore, the glacier's mass budget has shown a decreasing trend in (negative) magnitude between 2000 and 2020, primarily due to a reduction in ablation, especially in areas with thin debris cover and debris-free parts of the ablation area, which outweighs the relatively

smaller reduction in snowfall accumulation. More detailed findings and outcomes of the study are concluded as follows.

(1) The Batura Glacier exhibits substantial spatial heterogeneity in mass balance distribution along its elevation gradient. Altitudinal dependence was influenced by the presence of debris cover, resulting in the most intense melting occurring between 3000 and 3400 m, with a reversal of the ablation gradient below 3000 m due to the greater insulation by thicker debris on the lower portion of the glacier.

(2) Our simulations revealed that supraglacial debris cover exerted a notable influence on glacier mass balance. Including debris cover in the energy balance model led to a 45% reduction in the magnitude of the negative mass balance of the Batura Glacier (with debris: -0.26 m w.e. yr$^{-1}$, without debris: -0.48 m w.e. yr$^{-1}$). This reduction was particularly prominent during the ablation season, highlighting the significance of debris cover in mitigating glacier ablation.

(3) The role of debris cover in altering energy exchange was multifaceted. Debris cover enhances net radiation income by reducing albedo but also promotes thermal transfer, which warms the debris and leads to a higher rate of energy transfer to the atmosphere through longwave emission and sensible heat, thereby reducing available melt energy compared with bare ice. This intricate interplay modified the glacier's response to energy budgets, ultimately affecting its mass balance.

(4) Our investigation into the effects of debris thickness on temperature gradients within the debris layer reveals a fundamental connection between debris thickness and its influence on melt processes. Thicker debris layers engender reduced temperature gradients, leading to reduced latent heat available for glacier melting.

This study significantly advances our understanding of energy and mass interaction on debris-covered glaciers in the Karakoram. However, in addition to the previously discussed impact of ponds and ice-cliffs on ice ablation, future work should also address the evolution of supraglacial debris thickness and glacier dynamics. These factors exert a significant influence on the energy reaching the glacier surface (Compagno et al., 2022; Huo et al., 2021b). Finally, this paper identified that the mass balance of Batura Glacier became less negative in the period 2000-2020, most likely due to a decrease in air temperature over the same period. This result supports wider findings associated with the "Karakoram anomaly" and this phenomenon warrants further discussion and investigation.

**Declaration of competing interest**

The contact author has declared that none of the authors has any competing interests.

**Data/Code availability**

HAR dataset is available from Institute of Ecology Chair of Climatology website at https://www.klima.tu-berlin.de/index.php?show=daten_har2&lan=en. Meteorology and ablation observations. Glacier surface elevation difference of Wu et al. (2021) is available upon request from the authors, the elevation difference produced by Hugonnet et al. (2021), Shean et al. (2020), and Brun et al. (2017) are available at https://doi.org/10.6096/13., from National Snow and Ice Data Center (NSIDC) at https://nsidc.org/data/highmountainasia and from PANGAEA website at https://doi.pangaea.de/10.1594/PANGAEA.876545. The KGI datasets are available from the National Cryosphere Desert Data Center of China at https://doi.org/10.12072/ncdc.glacier.db2386.2022. The observations collected by this research are available upon reasonable request from the authors. The COSIPY used in this study is available on GitHub at https://github.com/cryotools/cosipy. The code developed for calculating energy and mass balance on supraglacial debris is available upon request from the authors. The coupled model will be publicly available once some technical issues are fixed.

**Author contribution**

Yu Zhu: Conceptualization, methodology, model development, writing original draft, writing review & editing. Shiyin Liu: Conceptualization, supervision, project administration, funding acquisition. Ben W. Brock: Supervision, model development, writing review & editing. Lide Tian: Supervision, project administration. Ying Yi: Validation, formal analysis, writing original draft. Fuming Xie: Investigation, visualization. Donghui Shangguan: Investigation. Yiyuan Shen: Formal analysis, visualization.

**Acknowledgments**

The authors acknowledge financial support from the National Natural Science Foundation of China (Nos. 42301154 and 42171129), the National Key R&D Program International Science and Technology Innovation Cooperation Project (No. 2023YFE0102800), and the Postdoctoral Research Foundation of Yunnan Province (No. C615300504038). The authors express gratitude to Water and Power Development Authority (WAPDA) for contributing their meteorological data and debris thickness observations. Special thanks to Professor Tobias Sauter and his team for open-sourcing the COSIPY model. We also thank an anonymous reviewers and Dr. Alexander Raphael Groos for helpful comments and suggestions on this manuscript.

**Appendix A Correction and downscaling of the model Inputs**

**A1 Adjusting of precipitation**

Numerous research endeavors have elucidated notable biases in precipitation observations within and in the vicinity of the Hunza river basin. For instance, Winiger et al. (2005) discovered a noteworthy discrepancy, with precipitation at altitudes surpassing 5000 m exhibiting sixfold or more intensity compared to lower altitudes, as

deduced from station observations. Similarly, Tahir et al. (2011) ascertained a dissimilarity between runoff and
observed precipitation, with Dainyor station recording a runoff of 750 mm/yr but a mere 100 mm/yr of observed
precipitation. This asymmetry was also discerned in the neighboring region (Immerzeel et al., 2009). To make a
more accurate precipitation input for the simulation, we consulted the method proposed by Wortmann et al. (2018)
to rectify the precipitation data. This method entails the calibration of precipitation through the calculation of the
calibration factor $f_c(H)$, as expressed by the following equation:
$$f_c(H) = (c-1)\exp\left\{-\left[\frac{P_{LR}}{(c-1)*100}\right]^2 * (H - H_{max})^2\right\} + 1 \tag{A1}$$

Where $c$ represents the calibration factor, $H_{max}$ represents the maximum elevation at which precipitation
occurs, $P_{LR}$ signifies the elevation correction factor for precipitation. These parameters are determined using the
linear relationship proposed by Immerzeel et al. (2012), and the range of values for the determination is derived
from existing studies. The linear relationship can be expressed as follows:
$$\begin{cases} P_T = P_{HAR} * \left[1 + \left(H - H_{ref}\right) * P_{LR} * 0.01\right] & H_{ref} < H < H_{max} \\ P_T = P_{HAR} * \left[1 + \left(\left(H_{max} - H_{ref}\right) + (H_{max} - H)\right) * P_{LR} * 0.01\right] & H > H_{max} \end{cases} \tag{A2}$$

Where $H_{ref}$ denotes the reference elevation, which corresponds to the elevation at which the observed
precipitation closely matches the actual precipitation. $P_{HAR}$ and $P_T$ represent HAR precipitation and calibrated
precipitation. We determined $H_{max}$ and $P_{LR}$ by approximating the calculated $P_r$ based on the water balance
equation (Eq. A3) (Figure A1), with the range of values for $H_{max}$ and $P_{LR}$ referencing the priori studies. In the
Eq.3, $ET$ uses MODIS evapotranspiration products, $R$ takes the runoff from the watershed outlet observation
station (Dainyor station), and TWS takes the average of GLDAS and GRACE solutions.
$$P_r - ET - R - TWS = 0 \tag{A3}$$


**Figure A1** Comparison between corrected precipitation and precipitation calculated by water balance equation.
**A2 Downscaling of the model inputs**
In order to achieve the desired level of precision for mass balance simulation on a glacier scale, this study

downscaled HAR reanalysis data from 10 km to 300 m by using statistical methods. Special attention was given to the impacts of topography, slope, and aspect on meteorological factors during this process. The SRTM DEM with a spatial resolution of approximately 30 meters was utilized to obtain topographic features. In order to effectively represent topographical features on a glacier scale while maintaining optimal computational efficiency during the energy balance simulation process, the target grid size was set at 10 times the SRTM DEM (~300 m).

Based on water balance at basin outlet, the precipitation was first calibrated using remote sensing data and station observations to obtain the altitude gradient and maximum precipitation altitude (Supplementary Methods). After calibration, the altitude gradient of precipitation throughout the Hunza river basin was determined to be 0.18%/m. The maximum precipitation altitude of the Batura glacier was 4900 m. Then, the precipitation was downscaled at a resolution of 300 m for the Batura glacier by applying the Eq.1 provided in the Supplementary. Incoming shortwave radiation was downscaled by using the radiative transfer equation (Eq.4) on sloping surfaces. The details in solving this equation can be found in publication of Ham (2005). The correlation coefficient of incoming shortwave radiation before and after downscaling is 0.91, with an RMSE of 26, indicating the parameterization-based downscaling enables a more refined representation of spatial characteristics while preserving the original characteristics and trends of the data.

$$R_{gs} = R_b \left( \frac{\cos(\phi) \cos(i) + \sin(\phi) \sin(i) \cos(\gamma - \alpha)}{\cos(\phi)} \right) + R_d \tag{4}$$

In the above equation, $R_d$ represents scattered radiation, which is solved using a modified Gompertz function that quantifies the relationship between horizontal total radiation ($R_{gh}$) and clear sky index (CI) (Wohlfahrt et al., 2016); CI is determined based on radiation duration, while $R_{gh}$ is initialized as $R_{gs}$; $R_b$ denotes direct incident radiation and is calculated by subtracting $R_d$ from $R_{gh}$; $\phi$ and $\gamma$ represent solar zenith angle and azimuth angle respectively, which can be obtained using parameterization schemes proposed by Wohlfahrt et al. (2008); $i$ denotes the angle between the slope and horizontal plane, while $\alpha$ represents the azimuth angle of the slope.

Temperature, relative humidity, wind speed, and air pressure were downscaled using altitude gradient obtained from HAR data. Cloud cover was downscaled refer to the scheme of ERA5 (Muñoz Sabater, 2019). Owing to the absence of meteorological observations required for computing altitude gradient, the altitude gradient over a broader region (Karakoram Mountains), which encompasses the study area, were determined using HAR data to minimize errors. The altitude gradient for 2 m air temperature was calculated to be -0.0054 °C/m, while that for 2 m wind speed was 0.00078 m*s$^{-1}$/m. The rate for 2 m relative humidity was 0.014 %/m, and that for atmospheric pressure was -0.044 hPa/m.

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
