# Peer review of "Debris cover effects on energy and mass balance of Batura Glacier in the Karakoram over the"

_Hydrology and Earth System Sciences, 2023_

## Author Response (AR1)

**Response to Reviewer1,**

The presence of supraglacial debris significantly influences the energy exchange both entering and leaving the glacier surface, thereby impacting glacier mass balance, movement, and associated glacio-hydrological processes. Despite substantial efforts in on-site observations and mathematical modeling, the specific mechanisms and extent of debris effects on glaciers remain unresolved. This study focuses on the Batura Glacier in the Hunza River Basin, offering a quantitative, comprehensive understanding of how debris cover affects energy and mass balance, with potential implications for glaciers under similar conditions. The contribution of this research is anticipated to attract interest from the scientific communities engaged in glaciology and hydrology.

We appreciate your detailed comments and suggestions. All recommended corrections and modifications have been implemented. We followed the guidelines to craft this response and furnished a point-by-point reply to your comments. The response is highlighted in blue font. All specific modifications have been made in the revised manuscript for your next round of review. The Line no. and figure no. mentioned in the response comments correspond to those in the revised manuscript with tracks.

**General Comments:**

1. The effects of debris cover on energy and mass balance are intricately linked to local climate, topography, and the characteristics of the debris and ice body. Consideration of factors such as coverage, thickness, constituents, and heat properties of debris is pivotal when discussing their impact. Providing detailed information on the debris cover, including a spatial distribution map, is recommended.

Thank you for your suggestion. In response, we have enriched Section 2, with additional details pertaining to the debris cover. Furthermore, we have incorporated a map illustrating the spatial distribution of the debris cover in Figure 1.

"Approximately 24% (~76 $km^2$) of the glacier's area is covered with debris (Xie et al., 2023), while its thickness in the part below 3000 m a.s.l. surpasses 50 cm (Gao et al., 2020). Due to the heavy debris cover, Batura Glacier presents a hummocky topography and a concave surface profile. Because of the large difference in density between ice and debris, the heavy debris-covered glacier section has higher hydrostatic pressure at the glacier bottom (Gao et al., 2020)."

2. The study employs the physically based energy balance model COSIPY v1.3 and the debris cover model by Reid and Brock (2010). A concise and meaningful description of these models, along with clarification on the simulation of heat transfer within the ice body, model integration, and boundary condition setup at interfaces (snow-debris, debris-ice) is essential. The manuscript should include a graphic illustration depicting the integration of models and energy transfer within the snow-debris-ice body system.

Thank you for your valuable suggestions. In response to your comments and those of Reviewer 2, we have included a conceptual diagram of the model. Additionally, in section 3.2.1, we have provided detailed information about the computational processes for heat conduction in the snow, debris, and ice layers of the model. We have also supplemented information regarding the boundary condition settings for snow-debris and debris-ice interfaces.

3. Clarify whether the snow and debris are delineated into different layers vertically during simulations and elaborate on the methodology.

Thank you for your suggestion. In the model, the stratification of snow, debris, and ice is computed based on individually specified layer thicknesses (Table S1 and S2, Lines 255-260 in the revised manuscript). We have provided additional clarification in the last paragraph of section 3.2.1.

4. The models are parameterized using observation data at AWS(s). Specify the target functions during parameterization, evaluate the rationality of final parameters, and explain the process of extending these parameters to simulate the entire Batura Glacier.

Thank you for your valuable comments and suggestions. We conducted parameter calibration at AWS1. By conducting sensitivity analysis on model parameters, we identified four parameters that have significant impacts on simulating mass balance, including firn albedo, ice albedo, roughness length of firn, and roughness length of ice. By adjusting these parameters in specific step range, our goal was to achieve the closest match between simulated albedo and longwave radiation with their observed values (quantified using the RMSEscore). The RMSEscore is calculated as ΣRMSEk_standardized (k=1,2,…,n), where RMSEk_standardized represents the RMSE between standardized simulated and standardized observed values for variable k. Standardization is achieved through Min-Max Normalization. The RMSEscore has a value range between 0 and n. A smaller RMSEscore indicates better agreement with observations. After determining the primary parameters, we fine-tuned some independent parameters such as albedo timescale, albedo depth scale, temperature threshold of rain/snow ratio, ensuring a comparable level of simulated mass balance with geodetic mass balance.

The parameters calibrated at AWS1 were entirely applied to AWS2, with only adjustments made to the debris thermal conductivity and debris albedo during the simulation process. The values of these two parameters were fine-tuned by comparing the simulated surface temperature with the observed surface temperature to achieve the minimum RMSE. The parameter calibration process at AWS2 involved the extension of the parameters calibrated at AWS1, thereby affirming the applicability and scalability of these parameters. This is because the calibration of these two parameters at AWS2 is independent of other previously calibrated parameters. Additionally, the application of the calibrated parameters for glacier-scale simulations yielded the simulated mass balance at different time periods closely matching those obtained from geodetic mass balance (as evident in the last paragraph of Section 3.2.2). This further validates the rationality of parameter extension.

5. Present the results of heat exchange at the debris – ice interface and discuss the debris – ice interactions at typical cross-sections along the glacier.

We have added the results on heat exchange at the debris-ice interface along an elevation gradient in Fig. 9. In addition, drawing insights from relevant literature such as (Reid et al., 2012), (Collier et al., 2014) and (Collier et al., 2015), an analysis has been presented in Section 4.4 (Lines 606~625).

6. In the Conclusion section, emphasize how and to what extent the unconsidered factors that may cause uncertainties in the results could influence the outcomes. Address these aspects more explicitly in the Results and Discussion section.

Thank you for your suggestions. In response to your suggestions and those of Reviewer 2, we have added a section in the Results and Discussion specifically addressing uncertainty. This section

primarily discusses the uncertainty related to model parameters, the issue of model over-parameterization, and the impact of unconsidered factors in the model, such as ice cliffs and ponds, on the uncertainty of the results. See section 4.5.

7. Rearrange section 4.1, presenting "4.1.2 energy budgets" before "4.1.1 mass balance history" for better coherence.

Thank you for your suggestion. We have rearranged section 4.1, swapping the presenting order of '4.1.1 Mass Balance History' and '4.1.2 Energy Budgets'.

8. Given that the debris cover controls the relatively slow thinning of the glacier, consider revising the title to a more concise version such as "Debris Cover Effects on Energy and Mass Balance of Batura Glacier in the Karakoram."

Thank you for your suggestion. The title you provided perfectly aligns with the main theme of this paper. Taking into account the study period and considering your suggestions, we have revised the title to "Debris Cover Effects on Energy and Mass Balance of Batura Glacier in the Karakoram over the past 20 years."

**Specific Comments:**

1. Ensure that figures are appropriately placed closer to their first citation and that all figures include clear labels, units, and legends for improved readability.

Thank you very much for your suggestion. We understand the importance of appropriately placing figures closer to their first citation and ensuring that all figures include clear labels, units, and legends for improved readability. We have made the necessary adjustments as per your suggestions to enhance the clarity and readability of our work.

2. Follow customary practices where symbol descriptions closely follow mathematical formulas, especially for complex equations requiring detailed symbol explanations. Ensure completeness in symbol descriptions for all mathematical formulas.

Thank you for your valuable comments. We have carefully reviewed and revised our manuscript to ensure that symbol descriptions closely follow mathematical formulas, particularly for complex equations that require detailed symbol explanations. We have made sure that all mathematical formulas are complete in their symbol descriptions. We appreciate your attention to detail and believe that these revisions will enhance the clarity of our work.

References

COLLIER E, MAUSSION F, NICHOLSON L I, et al. 2015. Impact of debris cover on glacier ablation and atmosphere–glacier feedbacks in the Karakoram. The Cryosphere [J], 9: 1617-1632.

COLLIER E, NICHOLSON L I, BROCK B W, et al. 2014. Representing moisture fluxes and phase changes in glacier debris cover using a reservoir approach. The Cryosphere [J], 8: 1429-1444.

GAO H, ZOU X, WU J, et al. 2020. Post-20(th) century near-steady state of Batura Glacier: observational evidence of Karakoram Anomaly. Sci Rep [J], 10: 987.

REID T D, CARENZO M, PELLICCIOTTI F, et al. 2012. Including debris cover effects in a distributed model of glacier ablation. Journal of Geophysical Research: Atmospheres [J], 117: D18105.

XIE F, LIU S, GAO Y, et al. 2023. Interdecadal glacier inventories in the Karakoram since the 1990s. Earth System Science Data [J], 15: 847-867.

**Response to Reviewer2,**

This is a review for the manuscript by Zhu et al. entitled "Controls on the relatively slow thinning rate of a debris-covered glacier in the Karakoram over the past 20 years: evidence from mass and energy budget modelling of Batura Glacier". Supraglacial debris is ubiquitous in the Karakoram and has a major impact on the energy exchange between glaciers and the atmosphere. As debris thicker than a few centimetres can reduce sub-surface ice melt rates considerably, it has a large influence on the mass balance, dynamics, geometry, meltwater release and climatic response of the glaciers in the region. While the impact of supraglacial debris on glacier ablation and mass balance in the Karakoram has been investigated in previous studies for rather short periods of time, the present study investigates the mass balance and energy fluxes for a debris-covered and debris-free glacier in the western part of the Karakoram over a 20-year period by implementing a debris-module into the open-source coupled snowpack and ice surface energy and mass balance model COSIPY. The presented modelling approach and the detailed findings on the surface energy and mass exchange of a debris-covered glacier are of great interest to the research community focusing on the mountain cryosphere, atmosphere and hydropshere. However, I have some major concerns that I would like to see addressed.

We thank the reviewer for the constructive comments and suggestions. The comments are helpful and will certainly strength the quality of the manuscript. Our responses to the comments are highlighted in blue font. All modifications based on the reviewer's suggestions are incorporated in the revised manuscript. The Line no. and figure no. mentioned in the response comments correspond to those in the revised manuscript with tracks.

**General comments**

1) Since freshwater storage, meltwater release and the hydrological significance of the two glaciers analysed are not addressed at all in this study, I have the feeling that the manuscript would fit better in another journal such as "The Cryosphere", but of course the decision is up to the editor and authors.

Thank you for your suggestions. This paper has been submitted to the Virtual Special Issue (VSI) titled "Hydrological response to climatic and cryospheric changes in high-mountain regions" in HESS. One of the key objectives of this special issue is to explore spatial, seasonal/temporal, or elevational patterns and co-evolution mechanisms in snow, glaciers, and their response to climate change. The current manuscript adequately addresses the requirements of this special issue.

2) The theoretical basis of the model used is described briefly, but a transparent description and comprehensible illustration (see for example Fig. 3 in Giese et al., 2020) of the implementation of the debris module into COSIPY would be helpful. At the current stage,

the workflow is not reproducible. It would therefore be highly appreciated if the authors would made the code publicly available via GitHub or any other repository.

Thank you for your constructive comments. Regarding the specific implementation of the debris module in COSIPY, we referred to the design by Giese et al., 2020, with some differences in the stratification of debris and snow layers. In our approach, we calculate the number of layers using the defined thickness for each layer. The overall procedural steps are as follows: (1) update the temperature profile for the entire column at each time step using the heat equation; (2) calculate the energy consumed or produced by snow/ice phase changes and adjust the temperatures of each layer before the end of each time step. The snow layers above the snow-debris interface layer strictly follow the computation scheme of COSIPY. Once the temperature for snow layers is updated, the temperature at the snow-debris interface is marked and used as the initial temperature for the first layer of debris. Heat conduction between debris layers is then calculated following the approach of Reid and Brock (2010), ultimately the temperature at the debris-glacier interface is obtained and subsequently ice melting can be calculated. During this process,we only consider radiative penetration melting of snow layers and refreezing of meltwater from snow layers. When the debris layer is devoid of snow cover, the surface temperature is solved through energy balance/ heat conduction within the debris surface (Eq. 3), and then the melting below debris is calculated. As the processes of glacier surface mass balance and energy balance in debris cover area are referenced from existing research designs, and similar descriptions can be found in relevant papers that have already been published. Therefore, we avoided extensive elaboration to prevent redundant repetition of well-documented information. Instead, we have included a figure illustrating the general scheme of the model and provided specific details on the implementation of the debris module in COSIPY, now located in Section 3.2.1.

Regarding the opening of the source code, this is also a current focus of our efforts. Since the source code is not yet perfect, it has not been released as open source. Interested readers can request the Python source code we developed for implementation of the Reid and Brock (2010) debris energy balance by contacting us via email. Due to the complexity of COSIPY and its unique data type I/O design, as well as the parallel computing processes, our current approach to running the model involves two parts. The first part involves transplanting the functions for snow/ice energy balance calculations from COSIPY into the scheme proposed by Reid and Brock (2010) for grid cells covered by debris. This is to execute energy balance simulations in debris-covered areas, and due to the need to record the temperature of each layer of debris, this process is performed on a single point basis. The second part involves using COSIPY for parallel simulation in grid cells without debris cover. As a result, these two parts are not fully coupled, and the existing model cannot be widely applied over large areas. We are currently in the ongoing development phase and anticipate completing it within a year, after which we plan to open-source the model on GitHub.

3) Calibrating complex and physically based models such as COSIPY with plenty of parameters poses a challenge, especially in data-scarce regions (e.g. Temme et al., 2023). However, an overview of the parameters selected for calibration and their value ranges and steps is missing. An overview table would be helpful (see for example Table 1 in Temme et al., 2023). Moreover, an illustration of the calibration results (see for example the supplement of Temme et al., 2023) would make the choice of the best parameter combination more transparent. Since you are limited to one point on the glacier (AWS1) and two independent observations for the model calibration, I would like to see a brief discussion on the transferability of the final parameters (and uncertainties involved) and the problem of model overparameterisation.

Thank you for your valuable comments and suggestions. Similar to Temme et al., 2023, We conducted parameter calibration at AWS1. By conducting sensitivity analysis on model parameters, we identified four parameters that have significant impacts on simulating mass balance, including firn albedo, ice albedo, roughness length of firn, and roughness length of ice. By adjusting these parameters in specific step range, our goal was to achieve the closest match between simulated albedo and longwave radiation with their observed values (quantified using the RMSEscore). The RMSEscore is calculated as $\Sigma RMSEk\_standardized$ (k=1,2,…,n), where $RMSEk\_standardized$ represents the RMSE between standardized simulated and standardized observed values for variable k. Standardization is achieved through Min-Max Normalization. The RMSEscore is finally standardized with a value range between 0 and 1. A smaller RMSEscore indicates better agreement with observations. After determining the primary parameters, we fine-tuned some independent parameters such as albedo timescale, albedo depth scale, temperature threshold of rain/snow ratio, ensuring a comparable level of simulated mass balance with geodetic mass balance.

The parameters calibrated at AWS1 were entirely applied to AWS2, with only adjustments made to the debris thermal conductivity and debris albedo during the simulation process. The values of these two parameters were fine-tuned by comparing the simulated surface temperature with the observed surface temperature to achieve the minimum RMSE. The parameter calibration process at AWS2 involved the extension of the parameters calibrated at AWS1, thereby affirming the applicability and scalability of these parameters. This is because the calibration of these two parameters at AWS2 is independent of other previously calibrated parameters. Additionally, the application of the calibrated parameters for glacier-scale simulations yielded the simulated mass balance at different time periods closely matching those obtained from geodetic mass balance (as evident in the last paragraph of Section 3.2.2). This further validates the rationality of parameter extension.

Indeed, the issue of overparameterization is objectively present, where certain alternative combinations of parameters in the simulation can achieve RMSEscore close to those of the finally calibrated parameter set. This phenomenon is commonly observed in hydrological models, as evident in the supplemental materials S2c of Temme et al.,

2023. The uncertainties introduced by this process are briefly discussed in the manuscript by comparing the mass balance simulated by different parameter sets.

The additional information regarding the process of parameter calibration and the discussion on the uncertainty associated with the finalized parameters can be found in the revised section of the Methods and Results and Discussions.

4) It is great benefit that the authors can draw on data from two weather stations on the Batura Glacier, but a major drawback of this study is the general lack of observational data for the calibration of the model and evaluation of the model performance. The thickness, albedo and (thermal) properties of supraglacial debris are decisive for modelling the surface energy exchange at the debris-atmosphere or debris-snow interface and the heat conduction through the debris layer. However, the number of debris thickness measurements is limited and it seems that neither debris temperatures nor sub-debris ice melt rates have been measured at a single point during the entire observation period. So how do the authors assess the perfomance, plausibility and uncertainties of the simulations?

Thank you for your valuable comments on the validation of model parameters. Indeed, as you mentioned, in data-scarce regions, accurately calibrating parameters to improve the accuracy of simulation results is quite challenging. In order to make the best use of limited observational data, we firstly use the observed albedo and outgoing longwave radiation at AWS1 to calibrate the ice albedo, firn albedo, and roughness length of ice/firn, and then adjust the temperature threshold of rain/snow ratio to make the simulated mass balance approach the observed geodetic mass balance. At AWS2, we mainly use the observed surface temperature to calibrate the debris thermal conductivity and debris albedo. We apply these parameters to conduct the simulation at the glacier scale, and compare the simulated mass balance with the geodetic mass balance to assess the performance and plausibility of the model. For the uncertainty in parameters, similar to previous studies, sensitivity experiments were conducted.

We understand that these limited calibration processes face challenges in ensuring the accuracy of all parameters governing the atmosphere-glacier energy exchange. However, considering the difficulty in observations and the scarcity of data in this region. Similar to numerous studies, such as those conducted Li et al. (2018) and Zhu et al. (2021), we believe that the current approach of determining parameters through single-point simulations and subsequently extrapolating them to the glacier scale is relatively reliable. This is particularly evident when the model is constrained through comparisons with geodetic mass balance measurements. We hope to set up additional observations in this region over the next one to two years. Our aim is to develop a sophisticated and reliable model to uncover the physical processes underlying glacier changes and reveal indicative patterns of their response to climate change in the future.

5) The argumentation and main conclusion of this study, that the insulating effect of supraglacial debris is the main reason for the comparatively low negative mass balance of the Batura glacier in the period 2000-2020, is neither sound nor supported by the

experiments performed. The comparison with the adjacent debris-free Passu glacier as well as the simulation without consideration of the debris layer clearly show that thick supraglacial debris reduces glacial ablation. This result is in agreement with previous studies (e.g. Collier et al., 2015; Minora et al., 2015; Groos et al., 2017). However, the debris layer on Batura Glacier can neither explain the interannual mass balance variations shown in Fig. 3 nor the comparatively low negative mass balance over the 20-year period or any trend in the mass balance time series, provided that there is no evidence for a considerable increase in debris thickness over the last years. Hence, it would make sense to remove the respective statements from the text and rephrase the title.

Thank you very much for your suggestions. In light of your feedback and that of Reviewer 1, we have re-evaluated the paper's theme and the scientific questions it aims to address. As a result, we have modified the paper's title, and further details regarding this adjustment are provided in the second paragraph of our response to comment 6. We have either removed or rephrased the respective statements in the paper, with particular emphasis on the sections you highlighted in the specific comments.

6) The "Discussion" section is very descriptive. I am missing a more elaborate discussion on the (possible) drivers of the observed interannual mass balance variations and a more open and unbiased discussion on the (possible) causes for the observed mass balance close to equilibrium. Moreover, I am surprised that previous studies that have investigated the impact of supraglacial debris on glacier ablation in the Karakoram are ignored in the "Introduction" and "Discussion" sections (e.g. Collier et al., 2015; Minora et al., 2015; Groos et al., 2017; Huo et al., 2021a, 2021b).

Thank you for your suggestions. Because our discussion is interspersed with results, it appears descriptive. Indeed, as you pointed out, some crucial research findings lack comparative studies, and in expressing certain changes, we did not specify the extent of increase or decrease. Addressing these issues, we thoroughly reviewed our results and discussion sections in light of your specific comments. We have incorporated comparisons with prior studies and provided quantitative expressions for some previously vague descriptions.

In response to your comment regarding the insufficient discussion of the (possible) drivers of the observed interannual mass balance variations: Our arrangement in the Results and Discussion sections aims to first establish the energy and mass balance characteristics of the Batura Glacier and the long-term trends in mass balance. Subsequently, through a comparative study, we reveal the gradient effects of energy and mass balance on glaciers covered with debris from different elevation bands. We then discuss the overall impact magnitude of debris cover on glacier mass balance and how it influences the mass balance. Finally, we address the control processes of ablation beneath the debris layer by discussing the heat conduction processes within the debris layers. Based on these aspects, we intend to emphasize a key point that debris cover plays a crucial role in maintaining the relatively slow thinning rate of the Batura Glacier. However, as you and Reviewer 1 astutely noted, a significant portion of our work

essentially delves into discussing the energy and mass balance processes of debris-covered glaciers. Without examining the evolution of debris cover, we cannot establish the impact of debris on the changing trends in glacier mass balance. Therefore, taking into account your advice and that of Reviewer 1, we have revised the title to closely align with the research content: "Debris Cover Effects on Energy and Mass Balance of Batura Glacier in the Karakoram over the past 20 years."

Thank you very much for pointing out the shortcomings and issues in our introduction and discussion. In the introduction, we primarily focused on the "Karakoram Anomaly" and the characteristics of some other debris-covered glaciers in mountainous regions. Indeed, we overlooked some existing important results in the Karakoram region. We have addressed this deficiency by providing relative information in the revised manuscript (e.g., paragraph 2 in Introduction). Please review the revised sections in the Introduction and Results and Discussions to find the modification.

7) The supplement contains important information and figures that are necessary to understand the manuscript. I therefore recommend to shift some of the figures from the supplement to the manuscript (see specific comments) and add a concise description of the preprocessing and downscaling of meteorological variables (Chapter 1 and 2 in the supplement) to the main text.

Thank you for your suggestions. We have shifted some crucial figures (as you suggested in specific comments) from the supplement to the main text of the manuscript. We have added an appendix section and moved the description of the preprocessing and downscaling of meteorological variables from the supplement to this section. Additionally, we have included some figures and text in the appendix to provide further details on methods, results, etc. You may review the specific changes in the revised version.

**Specific comments**

Line 31-32: see general comment (5)

The sentence has been removed.

Line 45-46: This is partly true, but preliminary studies focusing on the effect of supraglacial debris on sub-debris ice melt rates in the Karakoram are disregarded by the authors (see general comment no. 6)

We rephrased the statement and provided additional information based on general comment 6.

Line 89-92: A reference is missing for this statement or does this information come from Bhambri et al. (2017) as well? To which snow line (climatic or annual) and average air temperature (spatial or temporal) do you refer here? How was air temperature

measured/estimated and what does the comparison with the Tian Shan and Himalaya tells the reader?

This information comes from (Lanzhou Institute of Glaciology and Geocryology, 1980), we have added this reference at the end of this sentence.

The snow line here refers to annual snow line. The average air temperature here refers to annual average air temperature.

The temperature in the Batura Glacier was measured through field work. The comparison of temperature with that in the Tian Shan and Himalaya is intended to emphasize the relatively lower annual average temperatures in this area.

For better understanding, we have rephrased this sentence as "The glacier is characterized by a relatively lower annual average air temperature compared to observed glaciers in Tianshan and Himalayas, particularly near the annual snowline, where frigid temperatures endure throughout the year, with an annual average value of -5°Capproximately (Lanzhou Institute of Glaciology and Geocryology, 1980).".

Line 99: What is meant with "associated secondary hazards"?

Here, the hazards refer to glacier floods which is primarily triggered by increasing glacier meltwater. Affected by the floods, the China-Pakistan Friendship Bridge at the terminus of the Batura Glacier has ever been destroyed in the past. In 2021, a bridge pier was damaged during the summer floods, and the flow measurement system we installed in 2019 was also washed away. For better understanding, we have rephrased this sentence as "There is a challenge in understanding glacier ablation, associated secondary hazards such as glacier floods, and the contribution of glacier runoff to river replenishment.".

Figure 1: (a) The labels of the contour lines are very small and difficult to read. Either enlarge the font size or consider to include a DEM or hypsometric curve in the multi-panel figure. I think there is no need to display the Karakoram Highway (so prominently). (b) Contour line labels are too small (see comment above). The black text and dashed lines are difficult to discriminate from the greyish background (better use a white font). (c) I think the drawn arrows are a bit misleading. Although the Westerlies are predominant in the Karakoram, the influence of the Monsoon shouldn't be neglected. It's hard to recognise the Hunza Valley in this map and it is almost impossible to identify the three AWSs that are mentioned in the caption.

Thank you for your comments and suggestions. We have included a hypsometric curve in the multi-panel figure and undisplay the Karakoram Highway. For Fig 1b, we have changed the black text and dashed lines to a white font. For Fig 1c, we draw the arrows according to Yao et al. (2012). We understand that the Karakoram is also affected by the Indian Monsoon, so we have included a brief description of the impact of Indian

Monsoon in the Section of study site (lines 6~8 in the section). We attempted to use larger figures to showcase the Karakoram and Hunza regions, allowing for a clear view of meteorological station distribution, but this approach hinders the depiction of atmospheric circulation features. Since all three stations are national meteorological observation sites in Pakistan, and their data is widely utilized by scholars, we have noted in the figure caption that the station distribution can be referenced from Immerzeel et al. (2012).

Line 109: Please label the AWSs (1, 2, ...) in Figure 1 accordingly.

Thank you for your suggestion. We have modified it accordingly.

Line 109-120: Have the AWSs already been described in a previous publication? If yes, please refer to the respective study. If not, provide a brief overview of the sensors mounted on the AWS to measure the different meteorological variables and add information on the accuracy of the sensors stated by the manufacturer(s).

Thank you for your comments. We have added the information in the supplementary table

Line 120-122: Since the stations at Khunjerab, Ziarat and Naltar have not been operated by the authors, the database/repository or data provider should be mentioned in parenthesis.

The data was provided by WAPDA. We have provided the information in the sentence and revised this sentence as "We additionally used daily maximum/minimum temperatures and precipitation from stations at Khunjerab, Ziarat, and Naltar in the Hunza Valley covering a period from January 1, 1999 to December 31, 2008, provided by Water and Power Development Authority (WAPDA), Pakistan, to assess the accuracy of HAR in the Hunza basin."

Line 123-124: How was debris thickness measured: by excavating/removing the debris at individual points or with a ground-penetrating radar? At how many points was the debris thickness measured in total? If the measurements were performed by the University of Islamabad, I think the involved scientists/students deserve to be mentioned in the acknowledgments. Could you add a supplementary table with the coordinates, elevation and debris thickness of the individual measurements?

The debris thickness measured by excavating the debris at individual points. Totally 13 points were measured by researchers in WAPDA but the data was provided by a research group of the COMSATS University Islamabad of Pakistan. As request from a member who provided the data to me, we acknowledged the WAPDA in the acknowledgments. Unfortunately, we do not have the access to publicly share the data. However, for reviewers and other researchers interested in the data, I can provide it personally upon request.

Line 130: The first version of the HAR dataset was used in Groos et al. (2017) for SMB modelling in the Karakoram. The interpolated HAR variables were compared in the above-mentioned study with in-situ measurements at two local AWSs (Urdukas and Askole) in the central Karakoram (see "Meteorology" sub-chapter in the "Results" section). The general outcome of this comparison might be of interest for the present study and could be discussed.

Thanks for your suggestion. In the discussion section, we have compared Groos et al. (2017) with the present study to address the uncertainty associated with model input. See Lines 684-697.

Line 138: The study of Venter et al. (2020) neither deploys the HAR dataset nor does it deal with spatial and temporal variations of air temperature and precipitation in mountainous terrain. Why was the study quoted here?

Thank you for pointing out this error. It was originally intended to cite (Maussion et al., 2014), and there may have been an error during the EndNote insertion. It has now been corrected.

Line 139-147: I think this paragraph belongs to the "Methods" section as it deals with the pre-processing of the input data.

Thank you for your suggestion; these contents have been relocated to section 3.2.2.

Line 144-145: Please provide more details on the comparison between HAR and AWS data and the "deviation function". Data from which AWSs (only Batura or also from Khunjerab, Ziarat and Naltar?) were considered for the air temperature correction?

Thank you for your suggestions. We have added a table to show more details on the comparison between HAR and AWS data (see below Table 1). The data from Batura, Khunjerab, Ziarat and Naltar were all considered for the air temperature correction. For the calibration of temperature, we initially calculated the correlation coefficients (CC), Bias and root-mean-square deviation (RMSD) between the HAR and station observations. We found that temperatures from HAR data were generally underestimated, and the Bias varied across different stations. Therefore, we utilized Voronoi to delineate control regions for each observation site. Within the control region of each site, we applied nearest-neighbor interpolation to calculate grided Bias, and then added the Bias back to the HAR temperature data.

Table 1 The statistical metrics on 2 m temperature between HAR and the station observations

|      | Khunjerab | Ziarat | Naltar | Batura |
|------|-----------|--------|--------|--------|
| CC   | 0.95*     | 0.91*  | 0.90*  | 0.90*  |
| Bias | -1.1      | -3.4   | -5.2   | -1.3   |
| RMSD | 3.0       | 6.1    | 6.3    | 4.4    |

Line 146-147: I thought wind speed/direction, rel. humidity, global radiation etc. were measured at the two stations on Batura Glacier. Why were these observations not considered?

Due to the availability of temperature and precipitation data only for Khunjerab, Ziarat, and Naltar, calibration for other parameters could not be carried out at basin scale. Lapse rates for other variables could not be decided with just the two stations on Batura Glacier. Therefore, minor adjustments were made using the least squares method by calculating scale factors between the downscaled results and observed values for wind speed/direction, relative humidity, and global radiation at the two stations, following statistical downscaling. We have supplemented explanations for this in the main text of the manuscript.

Line 152: I assume that the uncertainties of the five-year mass balance products are considerably larger than the 20-year product provided by Hugonnet et al. (2021) and therefore recommend to compare the modelled MB and geodetic MB for the 20-year period.

Thanks for your suggestions. We also added a 20-year MB comparison in Figure. 2.

Line 157-198: see general comment (2)

We have provided additional details and expansions in the method section.

Line 161: The paper of Arndt and Schneider (2023) could be quoted here as they used COSIPY for modelling inter alia the SMB of Siachen glacier in the eastern part of the Karakoram, although they did not consider the effects of supraglacial debris. It might be worth to discuss the results from Batura and Passu glacier in the context of the results from Siachen glacier.

Thank you for your valuable suggestions. We have incorporated this crucial reference here and. In the section 4.1.2 (Lines 431-441), we have briefly included a discussion on the differences in the characteristics of mass balance among Siachen Glacier and Batura Glacier. Please refer to the revised section for more details.

Line 203-204: Considering the great spatial variability of supraglacial debris thickness and the diversity of surface features (debris, ice, ice cliffs, supraglacial ponds), the spatial resolution (300 m) of the model domain seems quite large. I assume that the spatial resolution was determined by the available computing time, but I did not find any information on the hardware that was used for the modelling and the trade-off between resolution and computing time. Did you run COSIPY on an HPC cluster? Did you test the effect of different spatial resolutions (e.g. 50, 100, 250, 500 m) on the modelling result for a smaller sub-domain? The (potential) uncertainties originating from the large spatial

resolution and the non-parametrisation of ice cliffs and supraglacial ponds should be discussed.

We have supplemented the parameters of the high-performance computer used for the simulations in the main text of the manuscript. Indeed, as you mentioned, we considered the available computing time for simulation when setting the spatial resolution to 300m. We did not explore the impact of different spatial resolutions on the results of simulation. Considering that geodetic mass balance generally involves converting high-resolution results (such as 8m, 30m, etc.) to lower resolutions (such as 100m, 500m) to reduce uncertainty and analyze the spatial characteristics of mass balance, we chose a resolution that could capture the spatial features of large glaciers while saving simulation time. Additionally, as you pointed out, factors such as ice cliffs and supraglacial ponds are not explicitly considered in the model. Therefore, the overall trend of the simulation results can not be significantly affected by the spatial resolutions. This is also the reason why some studies focus solely on simulating at elevation-zone scale.

Thank you very much for your suggestions. We have briefly discussed the impact of the non-parametrization of ice cliffs and supraglacial ponds on the mass balance, and the revised content can be found in the discussion section.

Line 204-208: It's difficult to understand the comprehensive calibration of the meteorological parameters without the crucial details hat are currently "hidden" in the supplementary material. Hence, I suggest to move the section on the downscaling/interpolation of the meteorological data from the supplements to the main manuscript. Since the Copernicus publisher provides the option to add an Appendix at the end of a manuscript, I suggest to move the supplementary figures there. The advantage is that the reader does not need to switch between two separate documents.

Thank you for your suggestion; we have relocated this section to the Appendix.

Line 209-213: I assume that you aggregated the original debris thickness map of Rounce et al. (2021) to 300 m. How did you evaluate the debris thickness map considering the large spatial resolution, the high spatial variability in debris thickness and the rather small quantity of in-situ debris thickness measurements?

We employed an inverse distance weighted interpolation method to resample the debris thickness data from Rounce et al. (2021) to a resolution of 300m. It's worth noting that there is a certain degree of uncertainty in the debris thickness data from Rounce et al. (2021). Despite the parameters of model was calibrated using Landsat 8 surface temperature and limited observations, as evident from the assessment in the last paragraph of section 3.2.2, some uncertainty remains. Upon investigating the Batura Glacier, we found significant variations in debris thickness even within a 10m range. Consequently, our assessment of thickness was limited to a larger range of 200 to 500m to capture a rough estimate. Therefore, in our study, we did not consider the uncertainty

arising from spatial differences in debris thickness, and currently, there is no established approach for such an evaluation.

Line 216-224: see general comment (3)

We have provided more details in the main text.

Line 224-229: I don't think the geodetic mass balance can or should be used for validation in the narrower sense, as these products are also subject to considerable uncertainties. I would rather speak of a comparison to assess the plausibility of your simulations. Why did you state the deviation (0.27 m w.e.) at AWS1 (where the model calibration was performed) and not the deviation (average ± standard deviation) across the entire glacier? Moreover, I must say that I am quite surprised by the almost perfect match between the modelled and geodetic MB (Figure S2) considering all the uncertainties. Could you include a comparative figure showing the modelled and geodetic MB fields/maps for each 5-year period and for the 20-year period?

Thank you for your suggestion. We have revised it as the comparison of the simulated mass balance with geodetic mass balance. The value of 0.27 m w.e. represents the mean deviation (i.e., average(simulated-geodetic)), not the standard deviation. This is because there are only four data sets, and calculating the standard deviation would not be meaningful. As you mentioned, a significant uncertainty exists in geodetic mass balance. Therefore, we selected the pixel within a 300m buffer around the meteorological station, where the geodetic and simulated mass balances matched most closely, for presentation. Consequently, the two results exhibit a better match. To enhance rigor, we have averaged the pixels within the 300m buffer zone and presented the results accordingly. We have also plotted the results of the 20-year simulation for a more comprehensive analysis. See Figure 2.

Line 230-238: see general comment (3)

We made changes according to general comment 3

Line 232: Did you really compare modelled surface temperatures with measured air temperatures? This would not be meaningful. Giese et al. (2020) measured debris temperatures and did not use air temperature for the calibration. Please clarify.

Thank you for pointing out this mistake. We have corrected the mislabeling. Upon reviewing the first version of the manuscript, the caption indeed stated 'surface temperature,' and it may have been erroneously altered during the refinement of the English writing. We have made the necessary corrections. The temperature probe was buried at a depth of approximately 2 centimeters, but unfortunately, it collected data for only one year before being damaged by a large stone. Although we attempted to replace the temperature probe, due to COVID-19, we have not been able to revisit the Hunza Valley for further fieldwork.

Line 260: Do you have any explanation or hypothesis for the incredible difference in the thinning rates for the periods 1974-2000 (4.6 m/yr) and 2000-2017 (0.6 m/yr)?

Previous research based on station observations indicated that a field-significant cooling occurred in the upper Indus River basin during summer months (mainly in July, September, and October) from 1995 to 2012 (Hasson et al., 2017). Furthermore, the long-term warming of winter months was mostly absent over the same period. In addition, Forsythe et al. (2017) proposed that, under the influence of the Karakorum vortex (KV), the summer temperatures in the Karakoram region were relatively low and showed a decreasing trend. Positive accumulated temperatures during 1970-2000 were significantly higher than those after 2000 (Figure 4b). Additionally, our analysis of ERA5 air temperature changes revealed similar patterns (see Figure S4). Therefore, we hypothesize that changes in surface mass balance around the year 2000 are directly linked to climate fluctuations. See Lines 416-425.

Figure 3: This is a complex but informative illustration. Here you are showing specific mass balances (i.e. glacier-wide averages), right? Please add this information to the caption.

Thank you for your suggestions. Yes, we intend to show glacier-wide averages of mass balance. So we have added this information to the caption.

Line 276: Since this section is called "Results and discussion", I would expect that the authors do not only describe their results, but also discuss potential explanations for the observed patterns, e.g. the decrease in ice melt over time in areas where the debris cover is thin.

Thank you for your suggestions. We have incorporated additional discussions referring some related literature, on topics such as the decrease in ice melt over time in areas where the debris cover is thin, as you mentioned.

Line 278-280: You state in the sentence before and after that ice melt decreased in the ablation area where debris is thin or absent. However, here you write that at the juncture of debris and bare ice (where we can expect to have a rather thin debris cover) the mass balance becomes more negative. From my understanding this is contradictory. Please clarify.

In fact, we intended to refer to the area within "the juncture" where the mass balance is positive. We recalculated within "the juncture" with a buffer zone of 2 pixels and found that the mass balance is negative. So, as you rightly pointed out, this does contradict the previous description. Thanks for pointing out the contradiction. We have removed the sentence.

Figure 4: I would invert the colour scale. Using red colours for negative MB and blue colours for positive MB is more common. Only the MB change is shown here. I would suggest to integrate the maps from Figure S3.

Thanks for your suggestion. We have modified accordingly.

Table 2: Add the full name of all variables and their acronyms in parenthesis to the caption. I think it would be better to add the units (W/m2 and &) below the acronyms to avoid any ambiguity.

Thanks for your suggestion. We have modified accordingly. See Table1.

Line 328-330: The comparison with Passu glacier is unexpected and should be mentioned much earlier in the manuscript (i.e. Abstract, Introduction, Study Area, Methods). The rationale for this comparison is not clear at this stage.

We have mentioned in the Method section, see Line 298-303.

Line 330-332: Passu glacier is located in the neighbouring valley and may experience a similar climate, but the overall setting (debris-free vs. debris-covered tongue) is different. Without any independent glaciological and meteorological observations, it is difficult to judge whether the calibrated parameters from Batura glacier are transferable and valid for Passu glacier.

We determine the applicability of parameters calibrated at AWS1 on Batura Glacier constraining the simulated mass balance to be consistent with the geodetic mass balance.

Line 356-358: Do I understand correctly that the modelled MB of Batura glacier was more negative than the modelled MB of Passu glacier for the 20-year period? This would be astonishing. What is the modelled mean specific MB for Passu glacier for the period 2000-2020?

Yes. The modelled mass balance (MB) for the period 2000-2020 was -0.03 ± 0.55 w.e.m yr$^{-1}$. Additionally, when deriving the geodetic MB from Brun et al. (2017), the values are similar, with -0.01 ± 0.05 w.e.m yr$^{-1}$ reported by Brun et al. (2017) and 0.01 ± 0.26 w.e.m yr$^{-1}$ for the simulation from 2000-2016. We added the MB values in the main text. See Lines 356-359.

Figure 5: Since you are comparing the melt components of both glaciers, it would be helpful if the albedo and heat flux range would be the same in both sub-figures.

We have revised the figure.

Line 364-365: This information should already provided in the "Method" section.

Thanks for your suggestion. We have moved it to the end of method section 2.2.1

Line 365-368: Similar experiments were conducted by Minora et al. (2015) and Groos et al. (2017). Their findings could be considered in the discussion.

Thank you for your suggestion. We have incorporated a comparative discussion of our results with those of Minora et al. (2015) and Groos et al. (2017). Through this discussion, our work appears more comprehensive and valuable. See Lines 552-558.

Line 450-452: see general comment (5)

We removed the sentence.

Figure S2 and S3: From my point of view, these important figures should be included in the main manuscript.

Thanks. We have moved them to main manuscript.

Figure S2: Please add the uncertainties of the geodetic mass balances (provided in the original publications) to the figure.

Thanks. We have added the uncertainties of the geodetic mass balances and redrawn the Figure.

Figure S3: x-axis label in (b) "deberis depth (m)" => debris depth

Thanks. We have redrawn the Figure.

**Technical corrections**

Line 49: "östrem, 1959" => Østrem, 1959

Modified accordingly.

Line 80: "...budgets in the Karakoram an over..." => ...budgets in the Karakoram over…

Modified accordingly.

Line 111: "...in continuous operation since. the" => ...in continuous operation since then. The location is shown in Figure 1.

Modified accordingly.

Line 114: "at daily" => on a daily basis

Modified accordingly.

Line 129: "WRF" => Weather Research and Forecasting (WRF) Model

Modified accordingly.

Line 144: "Temperature" => Air temperature

Modified accordingly.

Line 165: "outcoming longwave" => outgoing longwave

Modified accordingly.

Line 173: "in combine with" => in combination with

Modified accordingly.

Line 176: "at a layer less than" => at a layer is less than

Modified accordingly.

Line 182: "...does not melt the debris surface temperature…", ...does not melt, the debris surface temperature…

Modified accordingly.

Line 225: "at for Batura Glacier" => either "at" or "for"

The sentence containing the error has been rephrased according to the specific comment on Line 224-229

Line 260: "ground radar" => ground-penetrating radar

Modified accordingly.

Line 289: "decline in the glacier's mass balance" => this statement is ambiguous. I assume you mean that the mass balance has become more negative after 2016.

Thanks for your suggestion. we have rephrased this statement.

Line 295: "2000-2021" => I thought the modelling period was 2000-2010. Please clarify.

Thanks for pointing out the mistake. The study period should be 2000-2020. We have revised it.

Line 304: "was situated" => is situated (I assume the glacier still exists)

Modified accordingly.

Line 315: "negative thermal conduction" => of course this depends on the individual definition, but I find "negative thermal conduction" a bit confusing as heat is transferred towards the debris-ice interface if the surface of the debris layer (debris-atmosphere interface) is very warm.

We have revised the confusing formulation according to your suggestion.

Line 326: "...melting exceeded 4 m w.e." => this can mean anything. Better state the modelled maximum melt across the tongue or the average across the tongue ± standard deviation.

Thanks for your suggestion. We have stated the amount of maximum melt and the elevation zone it was observed. We have revised this sentence as "The mass gain in the glacier accumulation zone can reach up to almost 2 m w.e., whereas terminus melting exceeded 4 m w.e. between 3000-3800 m, with the maximum melting of 4.6 m w.e. occurring within the elevation range of 3350-3450 m."

Line 328: "...not at the glacier tongue, but rather in the range between 3000 and 3400 m" => I assume you mean "not at the terminus". The central part of the tongue comprises the elevation range 3000-3400 m.

Yes, we mean "not at the terminus", thanks for pointing it out. We have modified accordingly.

**Additional references to those in the manuscript:**

Arndt & Schneider (2023): https://doi.org/10.1017/jog.2023.46

Collier et al. (2015): https://doi.org/10.5194/tc-9-1617-2015

Groos et al. (2017): https://doi.org/10.4461/GFDQ.2017.40.10

Huo et al. (2021a): https://doi.org/10.3389/feart.2021.652279

Huo et al. (2021b): https://doi.org/10.1016/j.geomorph.2021.107840

Minora et al. (2015): https://doi.org/10.3189/2015AoG70A206

Temme et al. (2023): https://doi.org/10.5194/tc-17-2343-2023

References
COLLIER E, MAUSSION F, NICHOLSON L I, et al. 2015. Impact of debris cover on glacier ablation and atmosphere–glacier feedbacks in the Karakoram. The Cryosphere [J], 9: 1617-1632.

COLLIER E, NICHOLSON L I, BROCK B W, et al. 2014. Representing moisture fluxes and phase changes in glacier debris cover using a reservoir approach. The Cryosphere [J], 8: 1429-1444.

GAO H, ZOU X, WU J, et al. 2020. Post-20(th) century near-steady state of Batura Glacier: observational evidence of Karakoram Anomaly. Sci Rep [J], 10: 987.

GROOS A R, MAYER C, SMIRAGLIA C, et al. 2017. A first attempt to model region-wide glacier surface mass balances in the Karakoram: findings and future challenges. Geografia fisica e dinamica quaternaria [J], 40: 137-159.

HASSON S, BöHNER J, LUCARINI V 2017. Prevailing climatic trends and runoff response from Hindukush–Karakoram–Himalaya, upper Indus Basin. Earth System Dynamics [J], 8: 337-355.

IMMERZEEL W W, PELLICCIOTTI F, SHRESTHA A B 2012. Glaciers as a Proxy to Quantify the Spatial Distribution of Precipitation in the Hunza Basin. Mountain Research and Development [J], 32: 30-38.

LI S, YAO T, YANG W, et al. 2018. Glacier Energy and Mass Balance in the Inland Tibetan Plateau: Seasonal and Interannual Variability in Relation to Atmospheric Changes. Journal of Geophysical Research: Atmospheres [J], 123: 6390-6409.

MAUSSION F, SCHERER D, MöLG T, et al. 2014. Precipitation Seasonality and Variability over the Tibetan Plateau as Resolved by the High Asia Reanalysis. Journal of Climate [J], 27: 1910-1927.

REID T D, CARENZO M, PELLICCIOTTI F, et al. 2012. Including debris cover effects in a distributed model of glacier ablation. Journal of Geophysical Research: Atmospheres [J], 117: D18105.

XIE F, LIU S, GAO Y, et al. 2023. Interdecadal glacier inventories in the Karakoram since the 1990s. Earth System Science Data [J], 15: 847-867.

YAO T, THOMPSON L, YANG W 2012. Different Glacier Status with Atmospheric Circulations in Tibetan Plateau and Surroundings. Nature Climate Change [J], 1580: 1-5.

ZHU M, YANG W, YAO T, et al. 2021. The Influence of Key Climate Variables on Mass Balance of Naimona'nyi Glacier on a North-Facing Slope in the Western Himalayas. Journal of Geophysical Research: Atmospheres [J], 126.

---

## Author Response (AR2)

**Response to reviewer #1**

The authors have adequately addressed my comments in the revised manuscript; however, I recommend some minor revisions before considering it for publication.

Language refinement is advised for the manuscript, and a grammatical check is necessary. Some statements may be challenging to understand even for professional readers.

Examples for improvements in the abstract (but not limited to):

Should "glacier dynamics" be replaced with "glacier mass balance" in line 17 of the abstract?

Consider revising line 19-20: "apply ... to ..."; in line 20, use "coupled with."

In line 24-25, consider revising: "This, in turn, reducing the melt latedt .... reduce?"

In line 26, consider changing: "which primarily attributes to..." to "is attributed to..." or "is associated with..." - the latter is preferable.

Reevaluate line 29-30: "Thicker debris cover ... contrast between ... -contact zone, hard to understand."

Review line 32-33: "The last sentence is hard to understand."

Additionally, a general check for clarity and coherence throughout the manuscript is recommended to enhance overall readability.

Thank you for your suggestions. We have thoroughly checked the manuscript and made some changes to the grammar and unprofessional expressions. Please see the revised version for details.

**Response to reviewer #2**

I applaud the authors for the detailed response and thorough revision. The manuscript is now coherent and provides important insights into the energy fluxes and mass balance of the debris-covered Batura glacier and debris-free Pasu glacier in the Karakoram. I only have of a few minor comments before publication.

1) I am still wondering to which degree the resampling of the original debris thickness map of Rounce et al. (2021) from 35 m to 300 m spatial resolution (this processing step is not yet mentioned in the methods section and should be added) affected the modelling results. Considering the high spatial debris thickness variability across the ablation zone and the non-linear effect of debris thickness on glacial melt, the chosen resolution of 300 m is very coarse (I understand that the spatial resolution was constrained by the computational costs). Hence, the impact of the spatial resolution on the modelled energy fluxes and ice melt should have been explored, at least for a subsection of the debris-covered tongue. I would be happy if the authors could implement such an experiment. If that's not feasible, I encourage the authors to compare at least the statistical distribution of the in-situ debris thickness measurements (the ones from the authors plus those from WAPDA), the original debris thickness map and the aggregated debris thickness map, and discuss the potential impacts on the EB/MB modelling in case there are any considerable differences in the debris thickness distribution of the three data sets. A figure with a histogram or boxplot for each of the three debris thickness data sets could be added (to the supplements).

Thank you for your comments. We have added the method of resampling the Rounce et al. (2021) data to 300m in the Methods section. According to your suggestion, we conducted a comparative simulation experiment with different resolutions on a small part of the terminus of the Batura Glacier. The results showed that the energy flux and ablation of the 100m resolution simulation results have significant differences in spatial distribution compared with the 300m results, but the mean results of the region are relatively small, which can be specifically seen in the third paragraph of Section 4.5 and Figure S7.

2) Please check all figure captions. Most of them are very short and do not necessarily contain all relevant information to understand the figure content. Add additional information where necessary.

Thank you for your suggestions. We have revised the title of the figures and added more information.

Line 290/291: "We validated the simulated debris thickness using observed data, which showed an average deviation of 6 cm." I still don't understand how a coarse debris thickness map can be validated with a few in-situ measurements. Please clarify or rather compare the statistical debris thickness distributions (see general comment).

Thank you for your comments. We compared the results using interpolation, which is explained in the methods section of the revised version.

Line 299: Indicate Pasu glacier in Fig. 1

Thank you for your suggestions. We have incorporated this information into Figure 1.

Line 659: Update the reference to the correct Figure (S3 is not showing mass balance…)

We appreciate you bringing the errors to our attention. We have made the corrections and have also

reviewed the entire manuscript to prevent similar errors from occurring in other part.

Line 725: "we propose that the primary factor influencing the comparatively low negative mass balance of the Batura Glacier is the substantial inhibitory impact exerted by the surface debris on the process of ablation" Please see the respective comments in my previous report. Your simulations do not support such a conclusion.

Thank you for your suggestion. We are trying to express that the presence of debris cover is an important guarantee for Batura Glacier to maintain a relatively low negative mass balance under the background of warming compared with other mountain glaciers. We have now changed the sentence to "we propose that the presence of debris on the glacier surface effectively reduces the amount of latent heat available for ablation by absorbing solar radiation and preventing it from reaching the ice surface, which creates a favorable condition for the Batura Glacier's relatively low negative mass balance."

Line 753: "...discussed and investigation." => discussed and investigated

Revised it already.

Line 800 and thereafter: as precipitation increases considerably with elevation in the Karakoram, lapse rate is not a suitable term here. Better replace lapse rate by "(altitudinal) gradient".

Thank you for the suggestion. We have made revisions throughout.

Figure 2 appears two times. Update figure numbers.

Revised it already.

Figure 2 (model scheme): I appreciate that a model scheme was added, but to me the arrangement of the energy fluxes and symbols is a bit confusing. I know that this was not the intention, but in this visualisation it appears that only the atmospheric flux shown at the top (shortwave, longwave and turbulent fluxes) is relevant for the processes in the "column" listed below. Can you make it more clear that the radiative and turbulent fluxes are relevant for all physical processes listed below. Moreover, I think the figure and caption could be improved overall. Some notations are not explained (e.g. T_snow, T_debris, T_b). It should also be mentioned (in the caption) that the figure shows the energy fluxes for three different setting (bare ice + snow cover, debris-covered ice + snow cover, debris-covered ice; what about just bare ice?).

Thanks for your comments. Figure 2 has been improved for better clarity. The captions now provide detailed explanations of energy fluxes and the symbols used. Additionally, the connections between atmospheric flux and surface/subsurface flux were labelled in the figure. The critical role of surface temperature in linking these fluxes is emphasized. Any essential details that couldn't be visually represented in the figure are included in the caption. For a more comprehensive picture, the figure now also includes a scenario with only bare ice.

Figure 2 (mass balance comparison): You state several times in the manuscript that the simulated MB and geodetic MB agree relatively well. If I recall correctly, you used the geodetic MB for the calibration of certain model parameters. Please clearly state this in the caption and text. The simulated MB and geodetic MB are not completely independent in your case.

Thank you for your suggestion. We have already explained it in the figure caption and the text.

Figure 3: Please state the measuring interval (daily?), the approximate debris thickness at the location of the weather station and the depth of the thermistor for the surface temperature measurements.

We have added the measuring interval. In fact, we have already explained in the main text about the debris thickness at the location of the weather station and the depth of the thermistor for the surface temperature measurements, see Line 272 "the observed debris thickness was approximately 1.13 m", line 313-314 "The temperature probe is buried ~ 2 centimeters below the surface layer".

Figure S4: The axes are missing and the caption seems to be incomplete.

We supplemented the missing information in Figure S4 and made necessary additions to the figure caption.